# Surface Modifications of Medical Grade Stainless Steel

**Nusrat Sultana** [1,2], **Yuta Nishina** [1,2] and **Mohammed Zahedul Islam Nizami** [2,3,*]

1 Graduate School of Natural Science and Technology, Okayama University, 3-1-1 Tsushimanaka, Kita-Ku, Okayama 700-8530, Japan
2 Research Core for Interdisciplinary Sciences, Okayama University, 3-1-1 Tsushimanaka, Kita-Ku, Okayama 700-8530, Japan
3 Mineralized Tissue Biology, Forsyth Institute, 245 First Street, Cambridge, MA 02142, USA
* Correspondence: mnizami@forsyth.org

**Abstract:** Medical-grade stainless steel (MSS) is one of the most widely used materials for implantable devices in biomedical applications, including orthopedic stents, dental implants, cardiovascular stents, cranial fixations, and surgical suture materials. Implants are exposed to corrosive body fluids containing chlorides, proteins, and amino acids, resulting in corrosion, wear, toxicity, inflammation, infection, and failure. MSS-based materials exhibit improved corrosion and mechanical resistance and suppress the degradation and release of toxic metal ions. Although MSS is manufactured with a passivating metal oxide layer, its anti-corrosion performance against chlorides and chemicals in body fluids is insufficient. Implants require biocompatibility, bioactivity, hemocompatibility, and sustainability. Antimicrobial activity and sustained drug release are also crucial factors. Therefore, stainless steel with desirable multifunction is in great clinical demand. This comprehensive review summarizes recent advances in the surface modification of MSS-based implants and their biomedical applications, especially in dentistry.

**Keywords:** medical-grade stainless steel; surface coating; corrosion; biocompatibility; bioactivity

## 1. Introduction

Medical-grade stainless steel (MSS) is a specific type of steel with a strictly defined composition and low carbon content [1]; it is mainly used in biomedical applications [2–4]. In biomedical applications, four different MSS were found to be used, namely austenitic steel, ferritic steel, martensitic steel, and precipitation hardening steel. Again, they are distributed in several series of grades/families. However, types 304 (18%–20% Cr, 8%–10.5% Ni, 0.08% C, 2% Mn, 0.75% Si, 0.045% P, 0.03% S and 0.1% N) and 316/L (17% Cr, 12% Ni, 2.25% Mo, 2% Mn, 0.75% Si, 0.50% Cu, 0.10% N, 0.03% C, 0.025% P, 0.010% S, and 65.345% Fe) are the first and widely used MSS in biomedical application as part of treatment device [5]. Steel was replaced by other materials, including ceramics [6], metal alloys [7,8], plastics [9], and hybrid materials [10–12]. Metal implants have become standard in the medical field for the treatment of various physiological impediments, augmentations, and restorations [13–15]. Although several alloys satisfy the mechanical properties required for implants, few satisfy the clinical requirements of corrosion resistance and biocompatibility. To date, no metallic biomaterial complies with all mechanical and biological functions of the body.

MSS is widely used in biomedical especially in medical, dental, medicinal chemistry, and pharmaceuticals for their comprehensive corrosion resistance, biocompatibility, and high-temperature resistance [16]. Various applications of MSS in biomedical applications are shown in Figure 1. MSS is produced with a protective layer. However, when it is exposed to a corrosive body fluid environment (chloride, proteins, and amino acids), and at the same time low oxygen concentration in body fluid prevents to form of a stable oxide layer on corroded MSS, thus releasing toxic metal ions such as nickel, chromium and others [17–19]. While MSS has an undeniable importance in biomedical applications

nowadays, several studies have been conducted to overcome its shortcomings by altering or functionalizing its surface using various chemicals, biomaterials, and techniques. Surface modification is an effective method for adapting bioactive components to meet clinical requirements. Coating is one such method that cannot only increase corrosion resistance but also improve implant–tissue interactions, influence biological responses (i.e., bioactivity, cytocompatibility), and drug release [20]. MSS was traditionally used to fabricate orthopedic implants, especially for bone fixation (bone plates, screws, wires, mini plates), dental implants, spinal fixations, catheters, and cardiovascular (terminal, stent) applications [21].

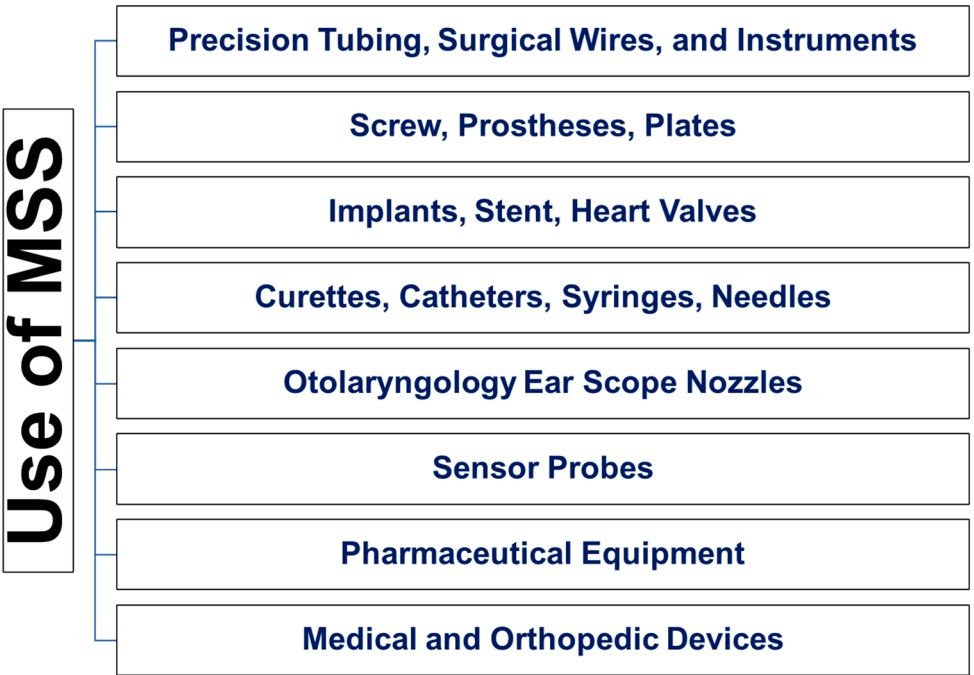

**Figure 1.** Use of MSS in biomedical applications.

Over the past few decades, surface modification of MSS to achieve optimal requirements was equivocal. Although MSS is biocompatible, its long-term corrosion resistance is questionable, and the exposure of the inner core and ion release are always a drawback. Alteration, functionalization, or modifications to the MSS surface allow it to improve its properties without compromising its sole properties. The existing studies on this topic have not been illustrated comprehensively in early published reviews. This review aims to provide an analysis of research conducted in this field and bring a complete summary of MMS surface modifications, used methodologies, and their outcome, as well as guide the selection of appropriate modifications for specific applications. At the same time, this review highlights research directions, recent developments, unresolved issues, and the feasibility of future studies in this field. Therefore, the maximum number of studies related to MSS surface modification for biomedical applications was included in this review but excluded abstracts, editorials, letters, and literature reviews.

Therefore, the EMBASE, MEDLINE, and Google Scholar databases were searched with keywords "medical grade stainless steel" and "AISI 316", "316L", "304", "SS", "1010", "implant", or "stent". Other related keywords, "surface coating" and "stainless steel" were also used for search refinement. These keywords covered as much information as possible about MSS in orthopedic, dental, cardiovascular, ocular, GIT, and urinary systems without overlooking relevant research. Among a large number of articles, several common ideas were found, including biocompatibility and bioactivity, osteointegration, corrosion resistance, the addition of antimicrobial activity, accretion of drug delivery systems, improvement of hemocompatibility, and improvement of physical, inflammatory, and other properties of MSS. The categorized applications of surface coating are summarized in separate tables for

each section, with their target applications presented schematically. An abbreviated form of the material names is used in tables and for general understanding, all types of medical grade stainless steel are denoted as "MSS".

## 2. Improvements of Properties of Bare Medical Grade Stainless Steel

Various modification methods were used to modify or enhance the surface characteristics of MSS for targeted applications and the development of enhanced properties. Table 1 summarizes the common strategies that were used for the enhancement of properties of bare MSS.

**Table 1.** Surface modifications of MSS using different modification techniques.

| Target Properties Improvement | Modification Techniques |
|---|---|
| *Antimicrobial and anti-biofilm activity* *Biocompatibility* *Corrosion and wear resistance* *Drug delivery* *Hemocompatibility* *Osseointegration, bioactivity, cell adhesion and proliferation, and new bone formation* *Physical, anti-inflammatory, and miscellaneous* | Atomic layer deposition (ALD) [22–24] Chemical vapor deposition (CVD) [25–28] Cold low-pressure gas plasma [29] Dip-coating [30,31] Electrodeposition (ED) [32–39] Hydrothermal crystallization method [40,41] Laser surface melting [42–44] Layer-by-layer coating [45,46] Magnetron sputtering [47–53] Matrix-assisted pulsed laser evaporation (MAPLE) [54,55] Microarc oxidation [56,57] Physical vapor deposition (PVD) [58–62] Plasma-spray [63–65] Sol–gel coating [30,43,48,66–73] Solvent casting [74,75] Spin coating [76–79] Spray coating [80–82] Ultrasonic spray [80–83] UV irradiation [84–87] |

### 2.1. Improvement of Antibacterial and Anti-Biofilm Activities

Implant-associated infection is one of the most undesirable problems, often leading to infected non-union, increased morbidity, and substantially worse outcomes with chronic infection. Implant surfaces in wounds are susceptible to colonization, proliferation, and biofilm formation by pathogenic bacteria [88]. After the introduction of an implant, an adapting layer composed of host-derived adhesins covers the implant surface, which promotes the adherence of planktonic bacteria and leads to the formation of an extracellular polysaccharide biofilm. Once a biofilm is formed, bacteria can easily escape the immune system and antibiotics [89]. These infections typically require multiple debridement surgeries, long-term systemic antibiotic therapy, or implant removal. Additional surgeries and therapeutics increase healthcare costs, as well as recurrence and failure rates [90]. Owing to the difficulty in treating implant-related infections, strategies intended to stop infection and biofilm formation during surgery and the immediate postoperative period may serve as more effective changes that can prevent these infections completely. Anti-infective biomaterials were increasingly used as adjunctive strategies to prevent implant infections and inhibit biofilm-forming microorganisms. Generally, local antibiotic delivery, including degradable sponges, injected biomaterials, and coating of antimicrobial molecules on implant surfaces, were used [91]. These materials vary in the degradation and elution profile of antimicrobials and might require prefabrication to attach antimicrobial molecules or coating materials to implants. Antimicrobial coatings have great potential in this context. A schematic diagram of the release of drug molecules from the surface-modified MSS is given in Figure 2. Table 2 summarizes the data showing the effect of antimicrobial surface coatings on MSS.

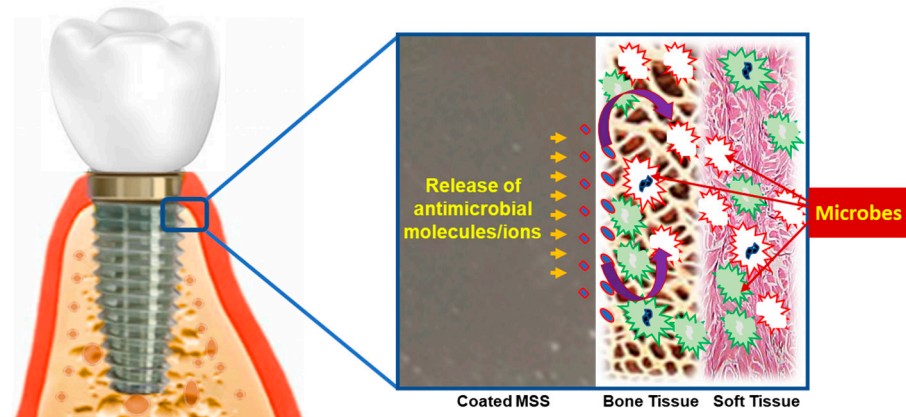

**Figure 2.** Release of drug molecules from the surface-modified MSS and their antimicrobial activity: Antimicrobial drug release occurs from the coated implant surfaces; Delivering the released drug to the tissue; The drug exhibits antimicrobial activities.

**Table 2.** Surface coating on MSS to improve its antimicrobial activity.

| Material (s) | Findings | Refs. |
|---|---|---|
| PC, Amikacin, VAN, Ti | Reduced bacterial growth and biofilm formation. | [91] |
| Cinnamon oil, CH | Reduced biofilm formation. | [92] |
| Particulate Ag | Reduced bacterial colonization and non-toxic bone cells. | [93] |
| Ag-HA-f-M, CNT | Reduced bacterial growth, enhanced corrosion resistance, and bioactivity. | [80] |
| Ag | Reduced intraoral biofilm and increased the bactericidal effect. | [94] |
| PBGHA | Enhanced bone-like apatite formation and provided an ideal surface for the stem cells' attachment and viability. | [75] |
| SAMs | Reduced biofilm formation. | [95] |
| Fluorine, Ag | Enhanced abrasion resistance and hydrophobic properties. | [96] |
| DLC | Showed similar bacterial adhesion like that on MSS. | [97] |
| Ag, PLGA | Reduced bacterial growth and enhanced osteoinductive properties. | [98] |
| CH, Bio glass-GEN | Reduced bacterial growth, and enhanced cell attachment and proliferation. | [99] |
| AgSiO$_x$C$_y$ | Reduced bacterial growth and enhanced biocompatibility. | [100] |
| Polyelectrolyte copolymers P, Polyallylamine hydrochloride | Reduced bacterial growth and enhanced anti-adhesion and cleanability. | [45] |
| NO | Reduced bacterial growth and adhesion. | [70] |
| Ag incorporated zeolite | Enhanced antibacterial activity and biocompatibility. | [40] |
| Ag-ZrO$_2$ | Enhanced antibacterial activity. | [101] |
| DLC, HA | Reduced biofilm formation and bacterial colonies. | [102] |
| Ag$^+$, AgCl, Cl$^-$ | Reduced bacterial growth. | [103] |
| AgCl, AgNO$_3$ | Reduced bacterial growth. | [104] |
| nZnO | Reduced bacterial growth and enhanced corrosion resistance. | [36] |
| Nano-Ag, Cu, Ti | Reduced bacterial growth. | [105] |
| Cu | Reduced bacterial growth. | [106] |
| AgNPs, AMP | Reduced bacterial and biofilm growth | [107] |
| PEM/AgSrMBG | Enhanced antibacterial activity, biocompatibility, bioactivity, and hemocompatibility. | [108] |

It was claimed that phosphatidylcholine (PC)-based materials could be loaded with antibiotics and applied as a coating on implants at the point of care, acting as an "antibiotic crayon". However, antibiotic elution and efficacy in inhibiting biofilm-based microorganisms were not yet characterized [91]. A study designed to determine the effect of coating on MSS implant surfaces using cinnamon oil and CH as bio-adhesives to prevent biofilms has demonstrated their efficacy [92]. A particulate Ag coating was used on MSS to prevent bacterial infection and was reported to be non-toxic to host cells in another study [93]. Ag-

substituted HA-functionalized multiwall carbon nanotubes (Ag-HA-f-MWCNTs) on MSS implants using spray pyrolysis were found to be antibacterial and corrosion-resistant [80]. In a separate study, Ag ions deposited on MSS using plasma immersion ion implantation showed increased bactericidal activity and biofilm reduction [80,94]. The PBGHA nanocomposite coating was prepared using a solvent casting process, was reported to be mechanically stable, and showed bioactivity due to the rapid formation of bone-like apatite on the coating. This provided an ideal surface for stem cell attachment and viability.

It was also reported to be an antibacterial coating [75]. Self-assembled monolayers (SAMs) were used to modify the surface of the MSS. Long alkyl chains terminated with hydrophobic (-CH$_3$) or hydrophilic oligo ethylene glycol (OEG) tail groups were used to form a coating via an orthogonal approach. SAMs were used to immobilize gentamicin (GEN) or vancomycin (VAN) on MSS to form an active antimicrobial coating that inhibits biofilm formation. GEN-linked and VAN-linked SAMs reduce biofilm formation [95]. MSS plates were modified using plasma-based fluorine and Ag ion deposition and this improved the abrasion resistance and hydrophobic property of MSS, which facilitated antimicrobial activity [96]. Bacterial adhesion to diamond-like carbon (DLC) is similar to adhesion to common MSS and its use on MSS or other materials without increasing the risk of implant-related infections was proposed [97]. Ag nanoparticle-PLGA-coated MSS alloy (SNPSA) is an antimicrobial implant material with cell proliferation and maturation properties [98]. A multifunctional composite CH-bio glass coating loaded with GEN-antibiotics was reported to improve the surface properties of metal implants. The coating supported the attachment and proliferation of cells and exhibited a bactericidal effect [99]. Protection of the implant surface against multidrug-resistant bacterial strains (MRSA) is also of great interest. In a study, an AgSiO$_x$C$_y$ plasma polymer coating exhibited antimicrobial activity against MRSA and biocompatibility [100]. MSS surfaces were embedded with several antimicrobial peptides in a multilayer film architecture using the layer-by-layer method, which exhibited antibacterial activity against both gram-positive and gram-negative bacteria [45]. The benefits of nitric oxide (NO)-releasing sol–gels as a potential antibacterial coating for orthopedic devices were also reported, whereby a coating of N-aminohexyl-N-aminopropyltrimethoxysilane and isobutyltrimethoxysilane for resealing NO was applied [70]. The functionalization of porous metals with antibacterial coatings was widely studied in recent decades. A highly porous MSS component was created by selective laser melting and an Ag-incorporated zeolite coating using in situ hydrothermal crystallization and was reported to inhibit bacterial growth and increase osteointegration [40]. In a further study, pulsed laser deposition was used to deposit Ag, ZrO$_2$, and Ag-ZrO$_2$ composite coatings on MSS, and their antimicrobial properties were reported [101]. It was reported that bacterial adhesion to DLC was similar to that of MSS [97]. However, in another study, DLC-coated pins prevented biofilm formation and bacterial colonization [102]. Other studies have reported that Ag-and chloride-electroplated MSS pins are bactericidal [103,104]. An organic/inorganic coating containing zinc oxide nanoparticles (nZnO) was obtained using the ED method on MSS and was reported to inhibit bacterial growth and increase corrosion resistance [36]. Ag-coated MSS showed a reduction in biofilm-forming bacteria compared with non-coated MSS or Ti implants [105]. Similarly, copper (Cu)-bearing MSS is antimicrobial and biocompatible [106]. A recent study reported that silver nanoparticles (AgNPs) and antimicrobial peptides (AMPs) conjugate developed a homogenous dispersion on MSS surfaces with moderate roughness and exhibited greater antimicrobial effects on *Staphylococcus aureus* and *Staphylococcus epidermidis* and their biofilm [107]. A functional polyelectrolyte multilayer (PEM) coating on MSS using spin coating was developed that included biocompatible and biodegradable collagen, γ-polyglutamic acid (γ-PGA), and chitosan. At the same time, mesoporous bioactive glass combined with silver and strontium (AgSrMBG) was also added. This multilayer PEM/AgSrMBG coating was reported to promote antibacterial activity, angiogenesis, and osseointegration [108].

Until now, various methods have been tested to improve the antibacterial and anti-biofilm activity of MSS surfaces. They mainly work to modify surface chemistry, energy, strength, wettability, and micro/nanostructure. In most cases, surface coatings with various organic or inorganic nano/coatings, including nanoparticles and/or drugs, as well as polymers and enzymes. Although they have shown reduced or inhibition of bacterial/biofilm adhesion, growth, and spread, they are largely subjected to insufficient mechanical stability and degradation, resulting in the long-term release of toxic components of MSS. In addition, some studies and illustrated results are not currently well-known in clinical settings. Furthermore, results and their ability to obtain antimicrobial/biofilm properties on MSS surfaces are sometimes not satisfactory. Therefore, alteration of surface morphology and sustainable coating is still a challenge and needs to be addressed.

### 2.2. Improvement of Biocompatibility

In terms of biocompatibility, pristine MSS usually exhibits weak interactions with body tissues. Therefore, coating with biocompatible materials is important for enhancing the biocompatibility of MSS (Figure 3). To develop the bio-interactive properties of MSS, various organic, inorganic, hybrid, polymer composite, and multilayer coatings were used.

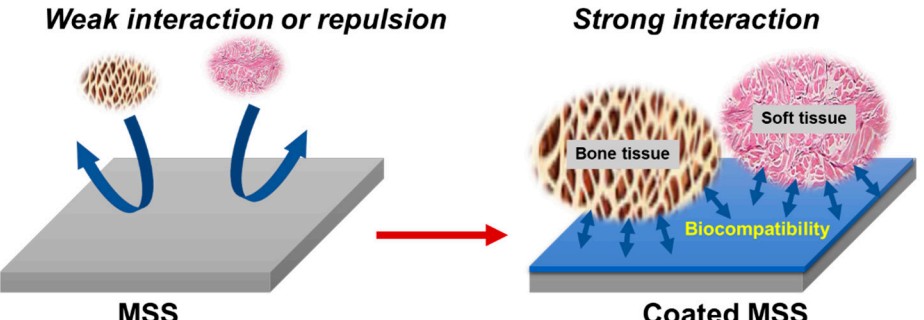

**Figure 3.** Surface modification of MSS to improve biocompatibility with bone and/or soft tissue: Bioinert stainless steel surface is not easily integrated with bone tissue and thus prevents bio interaction; Coated surface facilitates integration and prevents unwanted ion release, thus providing a biocompatible environment.

Table 3 summarizes the studies focusing on surface modification in terms of biocompatibility and bioactivity. The electropolymerization deposition of polyethylene glycol dimethacrylate (PEGDMA) on MSS surfaces was reported as a passive, hemocompatible, non-toxic, and durable coating. The coated MSS showed five times lower fibrinogen adsorption compared to bare MSS without platelet activation [109]. The use of hydroxyapatite (HA) and collagen as a bioactive coating material may enhance cell attachment, proliferation, and osseointegration. To immobilize HA-collagen without a high sintering temperature, a polydopamine (PDA) film was employed as an intermediate layer between the MSS and HA-collagen. These coatings enhance adhesion and surface roughness, and facilitate cell proliferation, alkaline phosphatase (ALP) production, and osteoblast attachment on the coated film [110].

In another study, MSS was laser surface melted using a continuous wave Nd-YAG laser in an argon atmosphere, resulting in an ideal crystallographic orientation on the MSS. This adaptation enhances cell proliferation and corrosion resistance [42]. The electrochemical oxidation of a Selenium (Se) coating on MSS with nano-pit arrays resulted in sustained drug release capabilities and enhanced cell adhesion, proliferation, and osteogenic activities [111]. Vascular endothelial growth factor (VEGF) R2 binding antibody fragments on MSS showed that immobilization of recombinant antibody fragments (scFv) facilitated human endothelial progenitor cell growth and cell viability of the implanted cardiovascular stents. However, scFv cannot be immobilized without prior aminosilanization of the surface [112]. However, HA on MSS showed strong bonding with surrounding bone

tissue, improved bioactivity, and osteointegration. Microarc oxidation of HA on MSS-Ti wires resulted in bone formation around the wires and increased pull-off strength [56]. Endothelial repair is a promising research area that may reduce stent restenosis and long-term post-implantation thrombosis. Human endothelial progenitor cells were identified as a key factor in re-endothelialization. At the same time, in a previous study, a novel coating of a combination of VEGF and anti-CD34 antibodies on coronary stents promoted the differentiation of human endothelial progenitor cells with low cytotoxicity [113]. In another study, chitosan (CH)-fluoride-doped diopside nanocomposite dip-coated MSS was found to be bioactive, corrosion resistant, and biocompatible [114]. PDA crosslinked chlorhexidine (CHX) on MSS was reported to be biocompatible with host cells. It exhibited cell differentiation, osteogenic maturation, and mineralization and provided a promising value for bone regeneration [115]. Inducing a mesoporous layer on the crystalline oxide facilitated cell activity and inhibited bacterial adhesion. $H_2SO_4/H_2O_2$ was used as an electrolyte for electrochemical oxidation to produce a thin crystalline mesoporous oxide layer on MSS. It was reported to influence the formation and maturation of focal adhesions, eliciting the outgrowth of filopodia and ultra-small lateral membrane protrusion effects on membrane fluidity. In addition, $H_2SO_4/H_2O_2$ facilitated nanoscale cell biomechanics and cell signaling. This effective layer on the MSS allows technology to shift away from the inherent limitations of traditional thick layers [116]. Using the matrix-assisted pulsed laser evaporation (MAPLE) method, a thin film of bioactive glass (BG61) or composite polymer-bioactive glass nanostructures (PMMA-BG61) was deposited on MSS. The nanostructured PMMA-BG61-MSS demonstrated good adhesion, durability, and peeling resistance, and served as an efficient shield against corrosion without affecting bioactivity and biocompatibility [54]. In a study, Fe-based steel electrode (Fe37Cr15Mo2B26C7Nb3Si3Al6Mn1) was deposited on MSS using the electro-spark deposition (ESD) method and reported to develop a uniform, dense, and optimum rough surface that shows biocompatibility and hemocompatibility [117]. Bioactive HA-titania ($TiO_2$) deposited and embedded in an MSS substrate using a combined laser–sol–gel technique was found to be bioactive and biocompatible [43]. The development of bilayer coatings using the electro-polymerization of poly(3,4-ethylenedioxythiophene) (PEDOT) on MSS followed by ED of strontium (Sr) and magnesium (Mg)-substituted porous HA (Sr, Mg-HA) was also found to be bioactive, biocompatible, and corrosion resistant [32]. Silicon-substituted HA nanoparticles combined with CH were coated on MSS by ultrasonication and found to have long-term biostability, bioactivity, and antimicrobial activity [83]. A poly (lactide-co-glycolide)-bioactive glass-HA (PBGHA) nanocomposite coating on MSS using a solvent casting process was reported to be bioactive and degradable without adhesive failure. The bioactive PBGHA nanocomposite coating was considered an ideal surface for stem cell attachment, viability, proliferation, and antibacterial properties [74]. A porous $TiO_2$-zirconia nanocomposite coated on MSS using a sol–gel process via dip-coating was reported to form one-dimensional rod-like carbonate-containing apatite and improve corrosion resistance. In addition, improved biocompatibility, cell viability, proliferation, and cellular attachment were demonstrated [30].

Graphene (G) deposition on MSS improved the adhesion and proliferation of human primary coronary artery endothelial cells and exhibited a unique potential on the endothelial cell phenotype by diminishing the endothelial-to-mesenchymal transition and enhancing the reduction of in-stent restenosis. In addition, it showed good biocompatibility [118]. A bilayer polypyrrole (PPy) coating on MSS using an ED method was reported, where a novel silica nanotube (SiNT) and ionic (such as Sr, Zn, or Mg) substituted HA composite (I-HA) were found to exhibit anti-corrosion properties, reduced rates of metal ions release, improved mechanical strength, enhanced bioactivity and apatite formation, as well as higher cell adhesion and proliferation [33]. Ti-C:H coating on bare, nitrided, and polished-nitrided MSS substrates using a closed field unbalanced magnetron sputtering system improved the wear resistance, corrosion resistance, and biocompatibility [47]. In another study, HA coating was formed by plasma spray on three austenitic MSS (ASTMF138, ASTM-F1586, and the nickel (Ni)-free Böhler-P558) and revealed that the three uncoated

MSS sheets and the HA-coated Böhler-P558 did not have any toxic effects on the cell. Although HA is biocompatible, the HA-coated ASTM-F138 and ASTM-F1586 showed high Ni elusions [63]. Therefore, it can be a harmful conjugation as well. Researchers must consider this issue. In another study, three different coatings [glow discharge nitrogen-doped (N-doped), carbon (C)-doped MSS coating sputtering, and low-temperature plasma N-doped] were used on MSS and demonstrated enhanced corrosion resistance, wear resistance, and microhardness. Although it was reported that ion-implanted and C-doped MSS were biocompatible, it was found that they affected cellular reactions when in contact with N-doped MSS. Although N-doped and C-doped layer depositions could be efficient options for improving the physical properties of MSS, questions remain concerning biocompatibility [119]. This biocompatibility issue is a significant issue for researchers to overcome to utilize the increased lifetime of MSS devices. The hybrid bioactive coating (TEOS-MTES-$SiO_2$) [(i.e., tetraethylorthosilane (TEOS), methyltriethoxysilane (MTES), and colloidal silica ($SiO_2$)] on MSS, applied using a sol–gel process, gave an idea about the bioactive-implant bone interface. Uncoated implants generate a thin bone layer at the beginning of the osseointegration process, but the layer becomes separated from the surface after a period. However, the hybrid coating generated new bone around implants, with a high concentration of Ca and P at the implant–tissue interface. The addition of bioactive silica nanoparticles enhanced the bone quality with a homogeneous Ca and P content, a low rate of beta carbonate substitution, and crystallinity-like young and mechanically resistant bone. The combination of glass-ceramic particles with a controlled rate of release of Si, Ca, and P played a significant role in bone formation around MSS implants [66]. A collagen-I coating on Ti and MSS implants using cold low-pressure gas plasma treatment showed an increased cell adhesion, cell growth, and biocompatibility of metal implants [29]. HA, Ti, and HA-Ti coating on MSS using plasma spray and the physical vapor deposition (PVD) process results in improved corrosion behavior [58]. The MSS surface was modified by a tantalum oxide ($Ta_2O_5$), followed by covalent coupling of fibrillar type-I collagen using a silane coupling agent (aminopropyl triethoxysilane) and a linker molecule (N, N'-disulphosuccinimidyl). The presence of collagen improved the cytocompatibility of the MSS implants. It was also assumed that the coating might serve as a depot for the release of tissue stimulants [i.e., transforming growth factor β1 (TGF-β1)] for better osseointegration [120]. In a study, $SiO_2$-$ZrO_2$ dip-coated MSS was more osteoinductive than $ZrO_2$ coating alone. However, the $ZrO_2$ coating was characterized as hydrophobic, while $SiO_2$-$ZrO_2$ was hydrophilic. The results suggested that the behavior of host cells in response to the biomaterial might vary depending on their origin [121]. Significant opportunities remain for researchers to undertake further biocompatibility studies using this type of coating.

**Table 3.** Surface coating on MSS to improve its biocompatibility and bioactivity.

| Material (s) | Findings | Refs. |
|---|---|---|
| PEGDMA | Developed homogenous, durable, and non-toxic coating. | [109] |
| PDA, HA-collagen | Enhanced adhesion, cell attachment, proliferation, and differentiation. | [110] |
| LSM | Enhanced corrosion resistance and cell proliferation. | [42] |
| Se | Enhanced cell adhesion, proliferation, osteogenic activity, and upregulated gene expression of OPN, RUNX-2, and ALP. | [111] |
| scFv, glycan-VEGF-$TiO_2$ | Enhanced immobilization of VEGFR2 binding recombinant antibody fragments without any toxicity. | [112] |
| HA, Ti | Enhanced bone formation and osteointegration. | [56] |
| VEGF, VEGF, Anti-CD34 | Enhanced re-endothelialization and reduced stent restenosis without toxicity. | [113] |
| CH, Fluoride | Enhanced corrosion resistance, bioactivity, and cytocompatibility. | [114] |
| CHX, PDA | Enhanced bioactivity, osteoblastic maturation, and mineralization. | [115] |

**Table 3.** *Cont.*

| Material (s) | Findings | Refs. |
|---|---|---|
| $H_2SO_4$, $H_2O_2$ | Enhanced cell activity and sensing filopodia, and reduced bacterial adhesion. | [116] |
| BG61, PMMA-BG61, | Enhanced bioactivity and corrosion resistance. | [54] |
| Nd: YAG laser, HA, $TiO_2$ | Enhanced biomimetic apatite formation and biocompatibility. | [43] |
| PEDOT, Sr, Mg-HA | Enhanced adhesion strength and bioactivity. | [32] |
| HA, CH | Enhanced corrosion resistance, anti-bacterial activities, and apatite formation. | [83] |
| PBGHA | Enhanced bone-like apatite formation, stem cell attachment, and viability. | [74] |
| $TiO_2$, $ZrO_2$ | Enhanced corrosion resistance and biocompatibility. | [30] |
| Graphene | Enhanced endothelial cell phenotype and endothelial-to-mesenchymal transition. | [118] |
| Smart ion (Sr, Zn, Mg), HA, SiNTs, PPy | Enhanced corrosion resistance and osteoblast cell attachment. | [33] |
| Ti-C:H | Enhanced wear resistance, corrosion resistance, and biocompatibility. | [47] |
| HA | Reduced cytotoxicity. | [63] |
| Fe- based metallic glass | Reduced cytotoxicity and enhanced cell attachment | [117] |
| N-doped, C-doped, Plasma nitriding | Enhanced physical properties without affecting biocompatibility. | [119] |
| Glass-ceramic-silica | Generated new bone around implants. | [66] |
| Collagen-I, Ti | Enhanced cell viability and cell attachment rate. | [29] |
| HA, Ti | Enhanced corrosion resistance. | [58] |
| Ta, $Ta_2O_5$, Collagen-I | Enhanced cell adhesion and proliferation. | [120] |
| $SiO_2$, $ZrO_2$ | Enhanced bone marrow-derived MSCs proliferation. | [121] |
| GO | Enhanced stability, non-reactivity, non-toxicity, cell adhesion, spreading, and proliferation. | [122] |
| Ferroelectric $LiTaO_3$ | Enhanced tissue regeneration and integration of the implant in the host tissue. | [84] |
| Ti-6A-l4V, $TiO_2$, $SiO_2$ | Reduced the production of proinflammatory cytokines by local tissues. | [67] |
| Silica, GlcNAc, Gal | Controlled glycan density. | [22] |
| $Ca_3(PO_4)_2$ | Reduced inflammatory response and enhanced biocompatibility. | [68] |
| HA, TiN | Formed HA coating | [123] |
| Hard $Cr_2O_3$ | Enhanced biocompatibility, corrosion, and wear resistance and showed less Cr ion release. | [48] |
| $ZrTiO_4$, $ZrTiO_4$-PMMA | Enhanced hydrophilicity, corrosion resistance, and cytocompatibility. | [69] |
| PPy, $Nb_2O_5$ | Enhanced biocompatibility and corrosion resistance. | [124] |
| $NbO_xN_y$ | Enhanced antibacterial activity and biocompatibility. | [125] |
| HA-zircon | Enhanced bioactivity, roughness, and hard tissue formation. | [126] |
| PPyNSE | Enhanced biocompatibility. | [127] |
| MAP | Enhanced biomolecule immobilization. | [128] |
| MPC. PHB | Enhanced biocompatibility and inhibit bacterial growth. | [129] |
| Nano-HA, Ni-P | Enhanced bioactivity and biocompatibility. | [34] |
| Sr incorporated $Nb_2O_5$ | Enhanced bioactivity, HA growth, and corrosion resistance. | [76] |
| PoP | Developed biocompatible carrier for vasoactive drugs. | [130] |
| TiN, NbN | Enhanced corrosion resistance. | [49] |
| HEP, Ta, Au | Reduced platelet activation and leukocyte–platelet aggregation. | [131] |
| Silicon | Reduced thrombogenicity and enhanced biocompatibility. | [132] |
| Graphene | Enhanced adhesion and collagen secretion of mesenchymal stem cells. | [25] |
| Ni-free MSS | Enhanced cell response and biocompatibility. | [133] |

Graphene oxide (GO) was immobilized on MSS by an amide linkage. It generated an adherent uniform coating with a surface roughness that improved hydrophilicity and biocompatibility by reducing the expression of reactive oxygen species [122]. In another study, MSS substrates functionalized with ferroelectric $LiTaO_3$ layers using electrical charging and UV light irradiation were reported to develop surface calcium phosphates ($Ca_3(PO_4)_2$) and protein adsorption. This study demonstrated the development of electrically functionalized platforms that can stimulate tissue regeneration and direct integration of the implant into the host tissue [84]. In another biocompatibility study, $TiO_2$ and $SiO_2$ coatings on MSS and Ti-6A-l4V using the sol–gel method reduced cytokine production, indicating that irrespective of the material used as a substrate, the reduction of the inflammatory response indicates

the improvement of biocompatibility [67]. An ultrathin $SiO_2$ coating on the MSS using ALD was used in a study and the MSS was functionalized with bioactive carbohydrates that are, N-acetyl-D-glucosamine (GlcNAc) and D-galactose (Gal), which contained a silane coupling reagent for linking to the surface. Surface-bound carbohydrates provided a new technique for the preparation of glycan-functionalized MSS and demonstrated the potential for the functionalization of MSS implants with bioactive carbohydrates [22]. $Ca_3(PO_4)_2$ coating on MSS using the sol–gel method showed a low inflammatory response, indicating an improvement in the biological acceptance of a conventional MSS implant [68]. Titanium nitride (TiN)-coated MSS was double-coated with HA by electrochemical post-deposition and was suggested to be promising because of the bone-implant interface with apatite. However, the findings were not clarified in detail in any biological experiments [123]. Another study showed that a hard chromium oxide ($Cr_2O_3$) coating on MSS using reactive magnetron sputtering demonstrated improved mechanical properties, including corrosion resistance, wear resistance, and hardness. The coated samples exhibited better biocompatibility. However, Cr ions were released during immersion tests [48]. Zirconium titanate ($ZrTiO_4$)-based sol–gel films were also suggested for improving the biocompatibility of metal implants. $ZrTiO_4$ and hybrid $ZrTiO_4$-PMMA thin films prepared by an aqueous particulate sol–gel method on MSS improve the hydrophilicity, cytocompatibility, and corrosion resistance of the substrate [69]. Electrochemical deposition of PPy and niobium pentoxide ($Nb_2O_5$) nanoparticles enhanced the mechanical properties, biocompatibility, and corrosion resistance of MSS [124]. In another study, niobium oxynitride (NbOxNy) coatings were coated on MSS using a reactive radio frequency magnetron sputtering technique found to be active antibacterial against S. aureus and E. coli bacteria. The coated substrates showed no toxicity on human fibroblast cells [125]. HA-zircon ($ZrSiO_4$) bio-composite coatings on MSS using the plasma spray method demonstrated bioactivity and roughness and were suggested to be suitable for biomedical implants. However, further studies are needed to validate these findings in clinical applications [126]. N-succinimidyl ester pyrrole (PyNSE) was electrocoated on MSS, followed by biomolecule [bovine serum albumin (BSA)] immobilization to PyNSE for a biomolecule-derivatized polymer coating, and evaluated for biocompatibility (thrombus formation, platelet adhesion, and hemolysis). The coating was reported to be more biocompatible and stable than bare metal [127]. MSS was coated using mussel adhesive protein (MAP), and then VEGF or CD34 antibodies were covalently immobilized on MAP to form a bio-functional film. The VEGF bio-functional film promoted the viability and proliferation of endothelial cells and the bioactive properties of CD34 antibody-coated stents. It was also reported that MAP coating allowed biomolecule immobilization, providing a promising platform for vascular device modification [128]. A study design coating of mussel-inspired surface attachable dopamine, lubricating zwitterionic polymers poly (2-methacryloxyethyl phosphorylcholine) (MPC), and a bacterial membrane destroying anti-bacteria molecule poly(3-hydroxybutyric acid) (PHB) on MSS surface reported biocompatible. They also demonstrated that its cell, platelet, and bacteria repelling inhibit bacterial growth [129]. Nano-HA and electroless Ni coatings were used as an interlayer on the MSS before the ED of the pure HA coating. The HA-incorporated nickel-phosphorus (Ni-P) interlayer showed biocompatibility in terms of forming a uniform pure HA layer when immersed in simulated body fluid. The HA layer suppressed the release of $Ni^{2+}$ ions through the interlayer, and these HA coatings also exhibited excellent adherence [34]. Another study showed that Sr-incorporated $Nb_2O_5$ coating on MSS using spin coating developed bioactivity and controlled release of Sr ions, which enhanced HA growth and corrosion protection [76].

Polymer coatings were suggested to decrease the thrombogenicity of metal intravascular stents. A biodegradable poly(organo)phosphazene (PoP) with amino acid ester side groups or a biostable polyurethane and butanediol coating were reported to induce a pronounced histiolymphocytic and fibromuscular reaction and functioned as a carrier for vasoactive drugs [130]. A DC-reactive magnetron sputter-deposited TiN–niobium nitride (NbN) multilayer on MSS was reported to be bioactive, anti-corrosive, and antibacterial [49].

In a study, heparin (HEP)-coated Ta stents and Au-coated stents induced less platelet activation and leukocyte–platelet aggregation [131]. In a clinical setting, MSS coated with hydrogen-rich amorphous silicon reduced thrombogenicity and improved biocompatibility [132]. In a study, "G" was developed on MSS using CVD and correlated with metallic Ni and $Cr_3C_2$ phases on the surface, and was found to promote the adhesion and collagen secretion of mesenchymal stem cells [25]. In another study, a high-N-Ni-free austenitic MSS was developed. The new Ni-free MSS exhibited high strength, good plasticity, cell response, and biocompatibility [133]. Various organics, inorganics, polymers, and hybrid composites were examined to improve the bioactivity and biocompatibility of stainless steel. While some of the results were outstanding, some results might have different compromising outcomes in response to coexisting properties. A realistic plan is yet to be performed to produce optimum biocompatibility of surface-coated or -modified MSS.

Based on the research available in this field, we can briefly say that the improvement of biocompatibility has a twist towards the advancement of biological characteristics of MSS. By employing appropriate changes to the MSS surface, cell adhesion, proliferation, and biocompatibility can be significantly modified and improved to maintain the stability, and functionality of MSS devices. Thus, it is necessary to develop strategies appropriate for the functioning of MSS surfaces by changing their composition and micro/nanostructure without compromising the original bio-mechanical properties.

### 2.3. Improvement of Corrosion and Wear Resistance

Corrosion is a vital aspect of biomaterials, especially for implant materials, in terms of durability, exposure of core metals, and the release of toxic ions (Figure 4), and leads to the failure of bioactivity, biocompatibility, and ultimately the failure of implant treatment. In the last few decades, the corrosion resistance of orthopedic stents [134,135], dental implants [136,137], cardiovascular stents [138,139], cranial fixations [140], and surgical medical sutures [141,142] was widely studied. Limited corrosion and mechanical resistance always lead to the degradation and release of toxic metal ions [143]. Although MSS is manufactured with a passive metal-oxide layer, its anti-corrosion performance against chloride and other chemicals in body fluids is not comprehensive. Therefore, the improvement of the anticorrosive capacity of MSS is a major research priority. Table 4 shows the different research strategies involved in corrosion resistance.

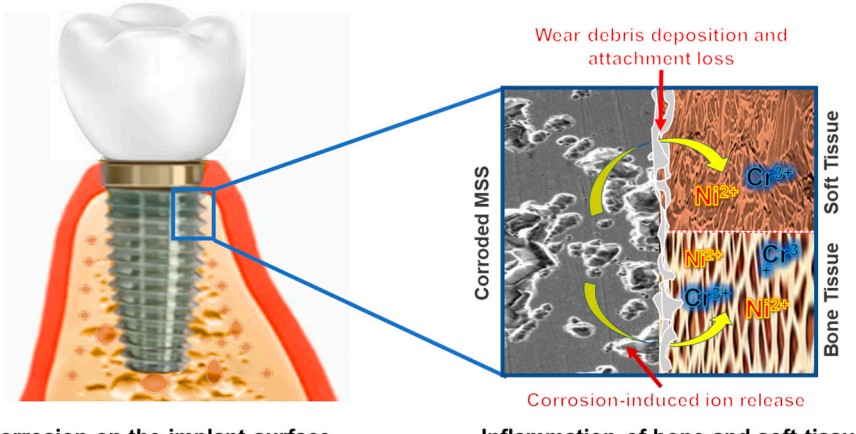

**Corrosion on the implant surface**          **Inflammation of bone and soft tissue**

**Figure 4.** Toxic ion release from corroded MSS; inflammatory changes of bone and soft tissue: Corrosion-induced toxicity leads to device degradation and toxic ion release; Wear debris gradually deposits to surrounding tissue and initiates attachment loss; Inflammatory response gradually proceeds tissue inflammation and implant loss.

**Table 4.** Surface coating on MSS to improve its quality of corrosion resistance.

| Material (s) | Findings | Refs. |
|---|---|---|
| Mn-HA, ZnO | Enhanced corrosion resistance and bioactivity. | [35] |
| TiN, SiO$_x$ | Enhanced failure of duplex coating. | [50] |
| HMDSZ, NIPAAm, AAc | Enhanced adhesion ability and corrosion resistance. | [85] |
| TiN, VN | Enhanced corrosion resistance and reduced bacterial attachment and colonization. | [51] |
| HfC, TaC, Au | Enhanced the micro-abrasive wear resistance and bioactivity. | [59] |
| PLGA, Ti-6A-l4V, Ti-6Al-7Nb | Reduced the degradation kinetics and enhanced the corrosion resistance. | [144] |
| Ti. HA | Enhanced corrosion resistance. | [145] |
| HA-ZnO | Enhanced corrosion resistance and inhibit bacteria. | [146] |
| TiN | Enhanced corrosion resistance. | [60] |
| Cr$_2$O$_3$ | Enhanced corrosion resistance and adhesion with a negligible chromium ion release. | [52] |
| Fluorocarbon | Enhanced corrosion resistance. | [147] |
| Ti, Ti-6Al-4V, Co-Cr-Mo, TiN | Enhanced mechanical properties. | [61] |
| TiN | Enhanced corrosion resistance and durability. | [148] |
| Diamond, Ti-6Al-4V, Co-Cr-Mo | Enhanced wear and corrosion resistance. | [149] |
| GO, Graphene-nanoplatelets | Enhanced corrosion resistance. | [77] |
| Ni | Enhanced corrosion resistance, conductivity, and hydrophobicity. | [26] |
| rGO nanosheets, Al$_2$O$_3$, TiO$_2$ | Enhanced corrosion resistance. | [23] |
| Graphene | Enhanced corrosion resistance and electrical conductivity. | [150] |
| Epoxy graphene | Enhanced corrosion resistance, UV stability, and impact resistance. | [86] |
| Graphene-nanosheet | Enhanced corrosion resistance. | [31] |
| Graphene | Enhanced corrosion resistance. | [27] |
| GO, PPy, Nanoplatelets | Enhanced corrosion resistance. | [151] |
| High-N | Enhanced corrosion resistance and biocompatibility. | [152, 153] |
| Zirconia | Enhanced corrosion resistance and biocompatibility. | [154] |
| f-MWCNT, BCP | Enhanced corrosion resistance and bioactivity. | [155] |
| Zr$_{48}$Cu$_{36}$Al$_8$Ag$_8$ | Enhanced corrosion resistance, and electrochemical stability, reduced the growth of bacteria, and toxicity. | [53] |
| ASP, N, Ag | Enhanced surface hardness, wear resistance, and antimicrobial activity. | [156] |

Porous manganese (Mn)-substituted HA (Mn-HA) coatings on zinc oxide (ZnO)-coated MSS using ED developed a uniform porous and strongly adherent coating that improved the corrosion resistance, mechanical properties, metal ion leach-out performance, bioactivity, and biocompatibility [35]. However, the tribocorrosion failure behavior was rarely studied. This exhibits the combined effects of wear and corrosion, which are usually caused by corrosion and transformations of substances. A TiN-SiO$x$ duplex coating was developed on MSS using plasma immersion ion implantation and deposition followed by radio frequency magnetron sputtering. This coating imparted a lower wear performance in NaCl, and synergistic wear-corrosion damage greatly accelerated the failure of the duplex coating. Although the TiN interlayer exhibited good adhesion, the SiO$x$ layer suffered from severe delamination during the sliding test in air [50]. In a separate study, an organic silicone film was coated on MSS using plasma deposition with a hexamethyldisilazane (HMDSZ) precursor. Ultraviolet (UV) light-induced graft polymerization of N-isopropyl acrylamide (NIPAAm) and acrylic acid (AAc) was applied to organic silicone to immobilize the thermos/pH-sensitive composite hydrogels. The coating improved the adhesion between the substrate and hydrogels and showed high corrosion resistance and drug release [85]. Nanoscale multilayered TiN-vanadium nitride (VN) on MSS using reactive DC magnetron sputtering exhibited wear and corrosion resistance [51]. In a separate study, tantalum carbide (TaC), hafnium carbide (HfC), and Au (TaC-HfC-Au) thin films on MSS using PVD improved wear resistance and bioactivity [59]. The polylactic-co-glycolic acid (PLGA) bioresorbable polymer coating on MSS, Ti-6A-l4V, and Ti-6Al-7Nb using the immersion method revealed that the PLGA polymer was degraded, but the composition of the copolymer remained unchanged, indicating that degradation did not change the

surface morphology and integrity of the underlying metal substrate and protected it from corrosion [144]. Cold rolling increased the hardness of MSS-Ti; however, deformation during cold rolling led to corrosion resistance. In this state, the HA coating improved corrosion resistance [145]. MSS was coated with HA-ZnO using ED reported to develop a uniform, corrosion resistant with low toxicity in a study. They also reported it to show antibacterial activity against *S. aureus* [146]. In a separate study, TiN ion-plated magnetic stainless steel (447J1) using PVD was found to develop durable corrosion resistance [60]. Moreover, a $Cr_2O_3$ coating on MSS using radiofrequency reactive magnetron sputtering was demonstrated to be corrosion-resistant, where $Cr_2O_3$ release was prominent without a trace of chromium ions [52]. In a study, a plasma fluorocarbon ultrathin coating on MSS was demonstrated to protect against MSS corrosion. Plasma etching with $H_2$ and $C_2F_6$ modified the chemical composition and thickness of the oxide layer and influenced subsequent polymerization [147]. In a study, the PVD technique was used to develop a TiN coating on four metallic substrates (pure Ti, Ti-6Al-4V, ASTM F138, and Co-Cr-Mo) and was reported to improve the corrosion resistance of MSS, but not of Ti and other alloys [61]. In another study, TiN films on MSS exhibited higher durability and lower corrosion [148]. Diamond has many superior desired characteristics of implant materials, including low friction, high wear and corrosion resistance, and good bonding to bone. In a study, an amorphous diamond coating was developed on MSS, Ti-6Al-4V, and Co-Cr-Mo and improved wear and corrosion resistance [149].

Promising applications of GO and graphene-nanoplatelets as corrosion inhibitors for MSS were reported. A coating was developed using spin coating and this exhibited corrosion resistance, pitting resistance, and a decreased passive current density [77]. In another study, graphene was deposited on MSS using CVD owing to the catalytic effect of the Ni-MSS double-layered structure. It was reported that a thin and multilayered graphene film continuously grew across the metal grain boundaries of Ni-MSS and enhanced corrosion resistance. Moreover, the formation of a passive oxidation layer on the MSS surface decreased conductivity [26]. A nanometric composite coating of laminate layers of Al and $TiO_2$ onto a thin layer of reduced graphene oxide (rGO) nanoplatelets using ALD on MSS revealed that rGO by itself did not provide protection, whereas the laminate reinforced the passivity by decreasing the passive current in which the rGO film acted as a primer for the anchoring of the ceramic layer [23]. A separate study demonstrated the growth of 3D networks of graphene-nano flakes on porous MSS substrates and reported enhanced specific surface area and increased corrosion resistance and electrical conductivity without altering the properties or the basic structure of MSS [150]. Spin-coated epoxy graphene (EG) nanocomposites on MSS substrates enhance corrosion resistance and UV stability [86]. When graphene-nanosheets were dispersed in 1-propanol and multilayer-coated on MSS using dip-coating it showed an increase in corrosion resistance [31]. Few-layer graphene with a low deflection could be grown on MSS foil using thermal CVD, which improved the surface roughness and surface area and reduced the corrosion current [27]. The use of carbon materials for corrosion resistance in MSS is interesting. Carbon black, carbon nanofiber, and carbon nanocages are mostly used as catalysts. A composite coating of GO and PPy on MSS was studied using an electrochemical galvanostatic deposition process, which showed improved corrosion and pitting resistance [151]. The S-phase (using plasma surface alloying with N, C, and both C and N) surface layers formed in MSS demonstrated that low-temperature nitriding, carburizing, and carbonitriding could improve corrosion resistance [152]. In addition, the attachment and proliferation of cells on the coated substrate were confirmed to be biocompatible [153]. A study developed a bioinert Zirconia coating Dysprosium phage stabilizing. They produced a uniform crack-free zirconia film over the MSS substrates by spin-coating. They reported that the coating acted as a physical barrier between body fluids and the implant surface and reduced corrosion. It was biocompatible with MG-63 cell lines and RBCs. At the same time, it showed the ability to form bone in the SBF [154]. A recent study developed a f-MWCNT)/biphasic calcium phosphate (BCP)

composites on MSS using ED technique and reported to achieve corrosion resistance and bone regeneration characteristics [155].

The MSS substrate was coated with thin-film metal glasses composed of $Zr_{48}Cu_{36}Al_8$-$Ag_8$ using magnetron sputtering and was found to exhibit corrosion resistance, electrochemical stability, antimicrobial activity, and no cytotoxicity to host cells [53]. Similarly, nanocrystalline Ag containing a wear-resistant S-phase was created on MSS using active screen plasma (ASP) alloying technology and was shown to improve wear resistance and reduce microbial growth [156]. Multidirectional research aimed at similar goals, such as corrosion and wear resistance, was conducted by different researchers. Although remarkable improvements were achieved, optimal goals are yet to be developed. Therefore, advanced research and their translation into clinical settings are necessary to improve surface modifications of MSS devices to facilitate corrosion and wear resistance.

### 2.4. Improvement of Drug Delivery Properties

Biofilms are a challenging problem for implant placement. One of the most effective strategies to combat biofilm-related infections is to stop adhesion and/or delay the growth, adhesion, and colonization of microorganisms. Local drug delivery is a promising method for preventing bacterial adherence and biofilm formation [157]. The implant surface should be coated with a bioactive coating, or the surface nanostructure of the metal implants should be modified for sustained release of antimicrobial drugs, ions, or chemical compounds [158,159]. Drugs can be loaded onto metamaterial surfaces by immobilizing chemical bonds [160] or by a passive coating that inhibits bacterial adhesion [161]. Multiple drug-eluting coatings, biodegradable components, therapeutic drugs, and biomolecules that could serve as multifunctional coatings for advanced implants were explored [162]. Local drug delivery can be used in implants. An illustration of this is shown in Figure 5. Sustained drug release could lead to the development of biofilm-free implants and tissue integration. Multiple research directions using several bio compounds are summarized in Table 5 to outline the advances in research on the development of drug-eluting implants. However, advanced research is yet to be conducted to develop an optimal coating for the desired outcome.

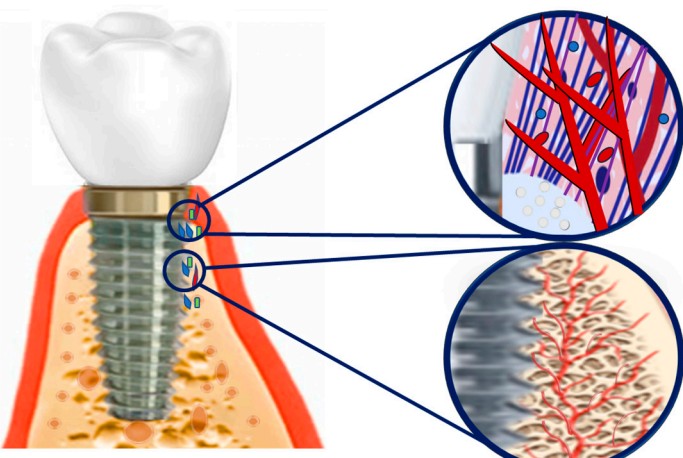

**Figure 5.** Drug delivery of drug-eluting modified MSS to bone and soft tissue for maintaining sound hard and soft tissue integrity: Release drug molecules; Provide a biocompatible and non-inflammatory environment; Maintain hard and soft tissue integrity.

**Table 5.** Surface coating on MSS to embrace drug delivery properties.

| Material (s) | Findings | Refs. |
|---|---|---|
| PCA | Enhanced wettability and surface area. | [87] |
| PEMs-57S | Reduced bacterial growth and maintained cell viability. | [78] |
| PMMA, Doxy | Reduced metal ion release, microbial growth, biofilm formation, and enhanced biocompatibility. | [55] |
| Ginseng, polyaniline, PLGA | Enhanced controlled drug release. | [37] |
| $TiO_2$ | Enhanced rapid endothelialization and reduced SMC proliferation. | [41] |
| DCF, CH | Enhanced controlled drug release. | [163] |
| F200, F202 | Reduced the adhesion/proliferation of host cells. | [164] |
| PPA, BuOPy | Enhanced biocompatibility and reduced toxicity. | [165] |
| $Al_2O_3$ | Reduced protein adsorption and platelet adhesion, and enhanced the attachment and proliferation of host cells. | [24] |
| PDA, PEI | Enhanced cell apoptosis and necrosis and anti-cancer function. | [166] |
| SZ-21, VEGF | Enhanced re-endothelialization and reduced thrombosis, inflammation, and in-stent restenosis | [167] |
| PLGA | Enhanced deformation and reduced the drug-eluting profile. | [168] |
| HUVECs, VEGF | Reduced neointimal hyperplasia and in-stent restenosis, enhanced endothelialization. | [169] |
| SiCOH plasma | Enhanced re-endothelialization and reduced in-stent restenosis. | [170] |
| SAMs | Enhanced drug delivery. | [171] |
| $Al_2O_3$, Tacrolimus | Reduced neointima formation and inflammatory response. | [172] |
| PLGA | Enhanced sustained-release profile with no significant burst releases, and anticoagulation behavior. | [173] |
| $Al_2O_3$, Tacrolimus | Enhanced antiproliferative effects. | [81] |
| LPPs, PLGA, type B gelatin | Reduced smooth muscle cell growth and enhanced healthy endothelium. | [174] |
| SAE | Reduced neointimal hyperplasia. | [175] |
| UL-MBCP, LPP | Enhanced non-viral gene delivery. | [176] |
| HA, $TiO_2$, Tobramycin | Enhanced fast-loading and controlled local drug administration. | [177] |
| Chondroitin sulfate, Px | Reduced neointima formation. | [178] |
| $Nb_2O_5$ | Enhanced bioactivity, controlled release of Sr ions, and corrosion resistance. | [79] |
| co-PEA, Tempamine | Enhanced biocompatibility. | [179] |
| $C_{22}H_{30}O_5$ | Reduced vascular macrophage infiltration and in-stent neointimal hyperplasia. | [180] |
| PTFEP | Enhanced biocompatibility. | [181] |
| PLA-DEX, PLA-SIM, PLA-PDLLA, PLA-PCL | Enhanced biocompatibility, reliability, and less neointimal hyperplasia. | [182] |
| PoP and APU | Enhanced biocompatibility. | [183] |
| Px, SMA | Enhanced vascular response | [184] |
| PAs, NO | Enhanced proliferation of endothelial cells, reduced proliferation of smooth muscle cells, and platelet attachment. | [185] |
| UL-MBCP, hMDp, Sirolimus | Enhanced mechanical properties, elution and degradation rates, and biocompatibility. | [186] |
| PTFE | Enhanced biocompatibility with no restenosis. | [187] |
| PTFEP | Enhanced thromboresistance and reduced late in-stent stenosis. | [188,189] |
| HEP | Enhanced corrosion resistance, reduced inflammation, thrombosis, and restenosis. | [190] |
| Endothelial cells | Enhanced therapeutic protein secretion. | [191] |
| Phosphorylcholine | Enhanced TIMP3 AdV transduction and transcription and reduced neointimal proliferation. | [192] |
| TiNOX, SS | Reduced MACE with no stent thrombosis | [193] |
| SPU | Reduced thrombus formation | [194] |
| HA, | Enhanced bone-pin interface | [64] |
| Phosphorylcholine | Reduced arterial neointima formation or luminal diameter | [195] |
| Carbofilm | Reduced stent thrombosis and restenosis. | [196] |
| PLL-g-PEG | Reduced neointimal hyperplasia and enhanced biocompatibility. | [197] |
| Au | Enhanced neointimal tissue proliferation. | [198] |
| Ag | Treated osteomyelitis without toxicity. | [199] |
| PEMs | Enhanced stent-mediated gene transfer. | [46] |
| Phosphorylcholine | Treated de novo coronary artery stenosis. | [200] |
| Porous MSS | Enhanced drug release. | [201] |
| EGCG | Reduced in-stent restenosis | [202] |

A poly(caffeic acid) (PCA) coating on MSS using UV irradiation under alkaline conditions was demonstrated to be bioactive [87]. Polyelectrolyte multilayer (PEM) coatings composed of collagen, CH, $\gamma$-poly glutamic acid, and tetracycline-loaded 57S mesoporous bioactive glass nanoparticles (57S MBG) were deposited on MSS using spin coating and have exhibited controlled release of tetracycline [78].

The MAPLE method was applied to print implants with doxycycline (Doxy) and polymer-bioactive glass systems and was demonstrated to exhibit prolonged release of Doxy and to develop bioactive HA in contact with body fluids. In addition, both polymer and apatite layers on the implant surface ensured protection against degradation and release of toxic metal ions [55]. In another study, a ginseng–polyaniline-encapsulated PLGA microcapsule coating was deposited on pre-treated MSS using ED and was reported to create a uniform microcapsule coating with low wettability [37]. A drug-free and polymerless surface on coronary stents using $TiO_2$ nanotexturing via the hydrothermal process was reported as a potential surface modification without the use of any polymers or drugs on MSS stents to overcome stent restenosis and thrombosis [41]. CH, a non-steroid anti-inflammatory drug (NSAID), and diclofenac (DCF) were used to prepare a drug delivery system using a multi-layered nanofilm system to make it more efficacious [163]. In another study, CH-derivatives were coated onto polyurethane and electropolished MSS to reduce thrombogenicity and platelet adhesion [164]. Non-biodegradable polymer coatings based on N-(2-carboxyethyl)pyrrole (PPA) and butyl ester of PPA (BuOPy) were electrodeposited on the MSS using cyclic voltammetry and created a thin uniform coating with various morphologies and hydrophobicities, which allowed paclitaxel (Px) loading and release [165]. An ultrathin layer of $Al_2O_3$ was deposited on MSS using ALD and modified with 3-aminopropyltriethoxysilane (APTS) and 2-methacryloyloxyethyl phosphorylcholine (MPC) to produce a phosphorylcholine-mimetic cell membrane surface, which inhibited protein adsorption and platelet adhesion, and promoted the attachment and proliferation of host cells [24]. A functional layer composed of PDA and polyethyleneimine (PEI) was fabricated on an esophageal stent. The PDA-PEI layer improved apoptosis and necrosis. The anti-cancer function of the PDA-PEI layers was correlated with the immobilized PEIs and suggested to be applied for the surface modification of esophageal stents [166]. In a study, a platelet membrane glycoprotein IIIa monoclonal antibody (SZ-21) and VEGF were loaded into the inner coating of MSS and modified with rapamycin (RAPA) and a drug-free PLA coating, demonstrating that the coated stents accelerated re-endothelialization and inhibited thrombosis, inflammation, and stent restenosis [167]. In addition, drug elution can be influenced by tuning the plasticizer and the PLGA/drug ratio [168]. The effects of gene transfection of endothelial cells with VEGF on re-endothelialization and in-stent restenosis inhibition were also reported [169]. SiCOH plasma nanocoating accelerates re-endothelialization and inhibits stent restenosis with less lumen reduction [170]. Ibuprofen-incorporated functional SAMs on MSS were reported as tethers for drug attachment and delivery from coronary artery stents [171]. A new inorganic ceramic nanoporous $Al_2O_3$ coating containing the immunosuppressive drug tacrolimus showed a reduction in neointima formation and inflammatory response [172,173]. Curcumin (CUR)-eluting PLGA coating on MSS stents using an ultrasonic spray was found to exhibit sustained release and improved anticoagulation behavior [81].

The therapeutic efficacy of endothelial-NO-synthase encoding plasmid DNA-administered stents was demonstrated through the inhibition of restenosis [174]. The effect of stent-based methotrexate delivery on neointimal hyperplasia was reported to be biocompatible and reduce neointimal hyperplasia [175]. A study reported that poly(beta-amino ester) (PbAE) pre-condensed plasmid DNA-containing cationic liposomes or lipopolyplexes (LPPs) immobilized on MSS using gelatin coating could act as a non-viral gene delivery system for the treatment of coronary restenosis [176]. A bioactive, anatase-dominated $TiO_2$ coating on MSS fixation pins using cathodic arc deposition showed antibiotic incorporation and sustained-release capacities [177]. In another study, chondroitin sulfate and gelatin-containing Px were reported for localized drug delivery to reduce neointima formation [178].

Sr-incorporated $Nb_2O_5$ was deposited on MSS using spin coating and was reported to be bioactive and corrosion-resistant [79]. MSS stents were dip-coated in biodegradable elastomeric poly(ester amide) (co-PEA), or polymer solution mixed with tempamine and were reported to decrease neointimal hyperplasia [179]. MSS coronary stents dip-coated in a biological polymer-methylprednisolone ($C_{22}H_{30}O_5$) solution were effective in decreasing both vascular macrophage infiltration and in-stent neointimal hyperplasia [180]. For local drug delivery, poly bis-trifluoroethoxy phosphazene (PTFEP) dip-coated stents have long-term biocompatibility as vehicles [181]. Self-expanding biodegradable PLA-dexamethasone (DEX) and PLA-simvastatin (SIM) with different coatings (PLA+PDLLA and PLA-polycaprolactone (PCL]) appear to be biocompatible, dependable, and reduced neointimal hyperplasia [182]. In a study, PoP-coated stents showed severe histiolymphocytic and fibromuscular reactions resembling a foreign body reaction [183]. In another study, the controlled release of Px from the stent coating of an elastomeric polymer mixed with styrene-maleic anhydride (SMA) copolymer demonstrated that Px was compatible with any component of the polymer blend [184]. A native endothelial extracellular matrix mimicking a self-assembled nanofibrous matrix was described as a new treatment model. The NO was released from the nanofibrous matrix, followed by a sustained release that enhanced the proliferation of endothelial cells, and decreased platelet attachment [185]. Urethane-linked multiblock copolymers (UL-MBCP) modified by fatty acid side chains were used for MSS coating with the anti-restenotic drug sirolimus as a biocompatible drug-eluting stent [186]. MSS stents coated with a polytetrafluoroethylene (PTFE) membrane were found to have an increased vascular lumen without any inflammatory vascular reaction or restenosis [187]. In addition, PTFEP was not found to have thrombus or restenosis on coated stents [188,189]. Polymer coatings on metal surfaces cause numerous problems after implantation, including late thrombosis, inflammation, and restenosis. In a study, HEP on different oxide films on MSS was found to be a potential substitute for polymer-coated drug-loaded stents to minimize corrosion, inflammation, late thrombosis, and restenosis [190].

Genetically engineered endothelial cells were seeded on stents using retroviral-mediated gene transfer and were demonstrated to secrete high levels of a therapeutic protein, which was suggested to improve stent function through localized anticoagulant, thrombolytic, or antiproliferative molecule delivery [191]. MSS stents coated with a high molecular mass polymer, phosphorylcholine, after treatment with recombinant replication-defective adenovirus designated as RAD TIMP-3, have the potential for the prevention of in-stent restenosis [192]. In another study, revascularization with titanium nitride oxide (TiNOX)-coated stents was suggested to be safe and effective for long-term treatment in patients with de novo native coronary artery lesions [193]. When an MSS stent strut was covered by a microporous elastomeric film minimal thrombus formation occurred [194]. In another study, MSS pins plasma-sprayed with HA showed bone coverage after implantation [64], while phosphorylcholine-coated metal stents were reported to not reduce restenosis [195]. In another study, a Carbostent (balloon expandable, MSS, a tubular stent with an innovative multicellular design and unique turbostratic carbon coating) showed no acute or subacute stent thrombosis [196]. The poly(L-lysine)graft-poly(ethylene glycol) (PLL-g-PEG)-adsorbed MSS stent surfaces showed low neointimal hyperplasia and reduced cell stent interactions and were reported to be biocompatible [197], while Au-coated stents have the potential to reduce neointimal tissue proliferation [198]. Electrolytically deposited Ag nanoparticles on MSS implants exhibit promising results in the treatment of osteomyelitis, without toxicity [199]. DNA-containing PEMs ionically cross-linked multilayers were fabricated layer-by-layer on the surfaces of balloon-mounted MSS stents using plasmid DNA and a hydrolytically degradable poly(β-amino ester) (polymer 1). Coated stents resulted in the expression of enhanced green fluorescent protein in the medial layers of the stented tissue and were suggested to be well-suited for stent-mediated gene transfer [46]. A biocompatible phosphorylcholine polymer-loaded MSS stent showed no major adverse cardiac events (MACE) or stent thrombosis [200]. In another study, porous MSS metal-based microneedle patches were reported to have potential biosensing and drug-delivery appli-

cations [201]. One study investigated mechanisms of epigallocatechin-3-gallate (EGCG) in HUVEC raised in MSS and reported significant inhibition of HUVEC proliferation and suggested that stent coating with EGCG had chemopreventive potential that could serve to treat in-stent restenosis [202].

Various types of drug-eluting surface modifications of MSS were tested. Typically, drugs are added to a coating used to functionalize the surface of the MSS for drug delivery. Thus, it is necessary to ensure that a therapeutic recommended drug concentration is maintained, otherwise, drug resistance or insufficient drug results will be shown. Therefore, sustainable, stimulus-responsive, and smart drug delivery systems should be developed with more advanced research in this field.

### 2.5. Improvement of Hemocompatibility

Many studies have used different coating techniques to improve stent restenosis. However, most have focused on the adsorption of platelets and albumin. Research is ongoing on the use of drug molecules and polymer coatings to resolve the problems of clot (thrombus) formation, platelet adhesion, thrombogenicity, and cyto and hemotoxicity by improving anti-adhesion properties, both adhesion and adsorption can be avoided, thereby overcoming thrombosis (Figure 6). Table 6 summarizes recent developments in the production of a hemocompatible surface coating on MSS.

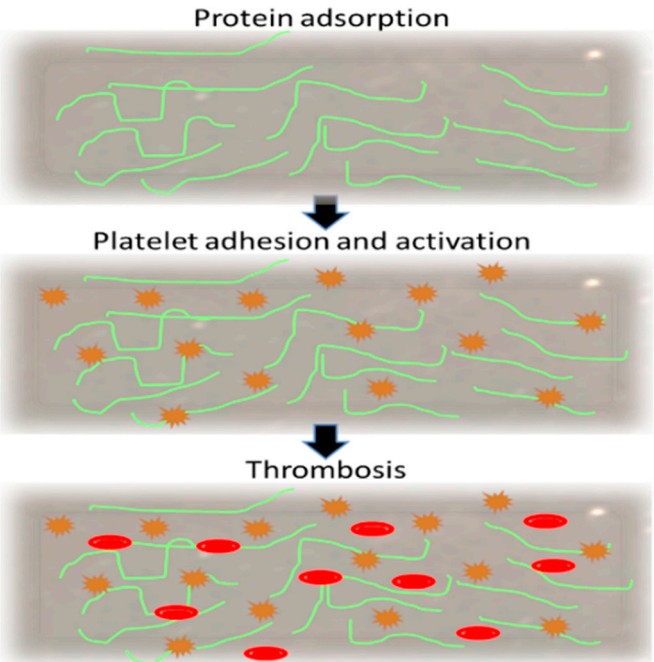

**Figure 6.** Schematic illustration of blood–implant interaction (without anti-adhesion coating): Protein adsorption initiates platelet adhesion, activation, and aggregation; Aggregated platelet leads to thrombus formation; Blocked off the stents.

**Table 6.** Surface coating on MSS to improve its quality of hemocompatibility.

| Material (s) | Findings | Refs. |
|---|---|---|
| PCL | Enhanced hydrophobicity, corrosion resistance, and anticoagulant properties. | [203] |
| PTFEP | Enhanced adherence of thrombocytes and hemocompatibility. | [204] |
| HEP, Poly-l-lysine microsphere, Dopa | Enhanced endothelialization and anticoagulation. | [205] |
| TiNOX, Ceramic | Reduced neointimal hyperplasia. | [206] |
| CH, dopa, PEG | Reduced platelet activation and clot formation. | [207] |
| PLA, PCL-PLA | Enhanced hemocompatibility, and reduced platelet deposition. | [208] |
| Gamma-APTS | Did not adsorb blood-clotting proteins or factors or stimulate them. | [209] |
| ChS, HEP, Au | Enhanced blood clotting time, and reduced platelet adhesion. | [210] |

**Table 6.** *Cont.*

| Material (s) | Findings | Refs. |
|---|---|---|
| GelMA-PEGDA | Enhanced endothelialization and anticoagulation. | [211] |
| BV, PDA | Enhanced aPTT and PT, reduced platelet, and fibrinogen activation. | [212] |
| Monocyte | Reduced Mac-1-mediated adhesion of monocytes. | [213] |
| HyA | Reduced platelet thrombus formation. | [214] |
| F202, Polyurethane | Reduced clot formation, platelet adhesion, thrombogenicity, and cytotoxicity. | [215] |
| RAPA, RAPA, CUR, PLGA | Enhanced hemocompatibility. | [216] |
| PEI, HN | Reduced platelet adhesion, and enhanced hemocompatibility. | [217] |
| PLA, HEP-P | Enhanced hemocompatibility. | [218] |
| EDOT, GO, PSS, HEP | Enhanced hemocompatibility. | [219] |

Stent thrombosis is an unresolved problem associated with the use of endovascular stents. Coronary stents prevent constrictive arterial remodeling and stimulate neointimal hyperplasia. Consequently, MSS surface coatings were assessed to reduce thrombogenicity. PCL film-modified MSS was found to improve hydrophobicity and lower BSA attachment, indicating improved anticoagulant properties. Additionally, it improved corrosion resistance [203]. Platelet adherence could be diminished by PTFEP coating on MSS stents. A previous study showed a reduction in the risk of acute or subacute stenosis following stent implantation, especially in microvascular obstructions, due to the release of platelet aggregates. However, it was also presumed that PTFEP coating had a favorable influence on restenosis [204]. A novel HEP-poly-l-lysine microsphere immobilized on a dopamine-coated MSS surface was shown to enhance antithrombin-III binding, partial thromboplastin time (PTT), and thrombin time (PT), suggesting its potential for coronary artery stent surface modification [205]. In another study, TiNOX coating on MSS reduced neointimal hyperplasia in MSS stents [206]. The MSS surface was coated with sulfonated CH using dopamine and PEG as anchors and was reported to limit platelet activation and clot formation, as well as calcium deposition [207]. The PCL-PLA-HEP coating enhanced hemocompatibility and reduced platelet deposition [208]. In a study, a small number of platelets adhered to alginic acid-immobilized MSS substrates [209].

Another study showed that a 5-layer chondroitin 6-sulfate (ChS)-HEP-modified MSS stent displayed the greatest hemocompatibility, prolonged blood clotting time, activated partial thrombin time (aPTT), and reduced platelet adhesion to reduce thrombosis [210]. In a study, methyl acrylate gelatin-polyethylene glycol diacrylate (GelMA-PEGDA) and polycaprolactone composite nanofibers composite loaded with rapamycin were sprayed step-by-step on MSS and reported to have improved anti-thrombosis and in-stent restenosis effects [211].

A hemocompatible surface prepared by immobilization of bivalirudin (BV) on MSS prolonged aPTT and PT and inhibited the activation of platelets and fibrinogen [212]. Adhesion of monocytes to the stent metal might contribute to the acute and chronic complications of stent placement. Based on the prominent electrochemical properties of the interaction between the monocyte integrin receptor Mac-1 and its various ligands, researchers have demonstrated that the Mac-1-mediated adhesion of monocytes to stents is inhibited by a silicon carbide coating [213]. In a study, the antithrombotic effects of hyaluronic acid (HyA)-coated MSS were shown to reduce platelet thrombus formation [214]. In another study, Hydromer's HEP-polymer complex (F202) was applied to a polyurethane film and electropolished on an MSS. It was found to form minimal or no thrombi on surfaces after exposure to recalcified human whole blood [215]. RAPA and RAPA-CUR-loaded PLGA coatings were fabricated on MSS stents using an ultrasonic atomization spray and were found to decrease platelet adhesion and activation, prolong aPTT clotting time, and decrease fibrinogen adsorption [216]. Bioactive nano-multilayer films consisting of PEI and hyaluronan (HN) prepared on MSS using electrostatic self-assembly were smooth, reducing platelet adhesion and hemocompatibility [216,217]. In another study, PLA and HEP-PCL-L-lactide-coated PLA appeared to be hemocompatible [218]. A hemocompatible coating

was developed using electrochemical copolymerization of 3,4-ethylenedioxythiophene (EDOT) with GO, polystyrene sulfonate (PSS), or HEP on MSS, which was found to produce an antifouling surface that prevented the restenosis of blood vessels and improved the anti-blood-clotting capability of cardiovascular stents [219].

Hemocompatible surface modification is a technique that involves understanding the blood reactions of biomaterials used. From the research, it is evident that functional biomaterials should be effective, biocompatible, and durable with high biomechanical resistance. To achieve desired treatment goals, such as platelet activation, anti-coagulation, in-stent stenosis, protein absorption, and antimicrobial infections, various surface functionalization techniques are being developed and used in clinical settings. Researchers are combining various surface functional materials with manifold activities to reach the ultimate goal. For instance, combining antimicrobial and anticoagulant surface coatings with bioactive proteins in MSS can increase hemocompatibility, as well as inhibit microbial infections and stenosis. Although several techniques are being used to develop composite products, more advanced research should be conducted to develop such a versatile coating for modifications in MSS.

### 2.6. Improvement of Osseointegration, Bioactivity, Cell Adhesion, Proliferation, and Bone Formation

The search results from studies that investigated improving osseointegration using various coatings on medical-grade stainless steel are summarized in Table 7. Surface modification of implants using biological coatings is thought to improve osseointegration. In addition, it reduces the risks associated with infections and corrosion [220]. In particular, bioceramics (i.e., bioactive glasses and HA) are known to functionalize metal implants [221]. However, the combination of bioceramics and biopolymer composite coatings has recently gained interest because of the organic/inorganic composition of natural bone [222]. These coatings also facilitate the incorporation of biomolecules, including antibiotics [223]. At this stage, single-to-multi-coating using versatile materials was used to enhance osteointegration. In Figure 7 the initiation of osteointegration by cell proliferation is represented.

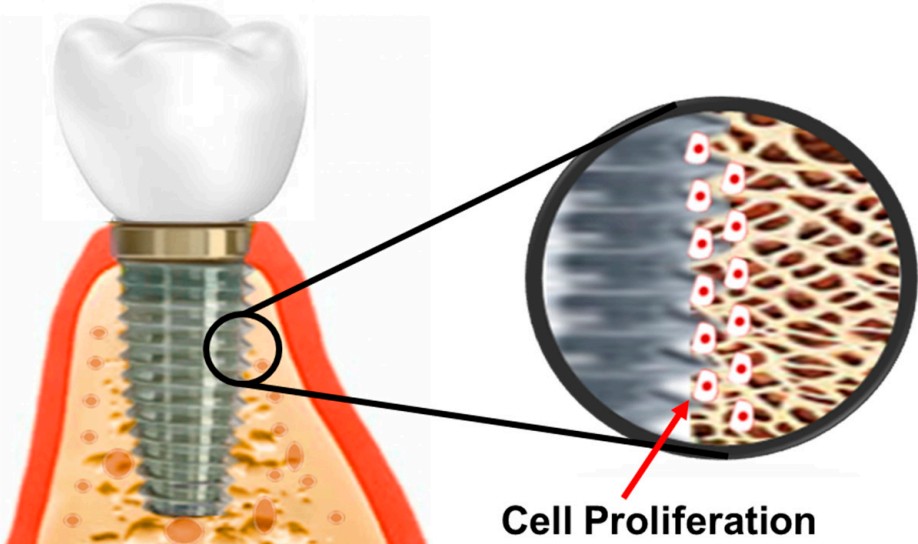

**Figure 7.** Surface-modified MSS facilitated cell proliferation and osteointegration: Release cell proliferative drug molecules; Initiate cell proliferation and differentiation; Facilitate osseointegration.

**Table 7.** Surface coating on MSS to improve osseointegration.

| Material (s) | Findings | Refs. |
|---|---|---|
| FN7-10 | Enhanced adhesion and differentiation of host cells. | [224] |
| PLLA | Enhanced osteogenic activities. | [225] |
| Ti | Enhanced the proliferation and migration of host cells, enhanced biocompatibility, and osseointegration. | [57] |
| HA, PMMA | Enhanced risk of infection. | [226] |
| sPGF, PAA matrix | Enhanced osteogenic differentiation. | [38] |
| Poly-OEGMA, Dopa | Reduced protein adsorption, cell adhesion, and enhanced osseointegration. | [227] |
| Bisphosphonates | Enhanced removal strength and bone remodeling. | [228] |
| Bisphosphonate, HA | Enhanced bone formation. | [229] |
| Bisphosphonate | Enhanced pullout force, increased pullout energy. | [230] |
| TEOS-MTES, HA | Enhanced bioactivity and implant–tissue integration. | [71] |
| HA, SS | Enhanced tight apposition between bone and the coating. | [231] |
| HA | Enhanced bone ingrowth, microhardness, and mineralization. | [232] |
| HA | Enhanced implant fixation. | [233] |
| TiAlN | Reduced osseointegration. | [234] |
| HA | Enhanced bone volume. | [235] |
| Femtosecond laser | Enhanced bone formation rate under osteogenic conditions. | [236] |
| HA, Ti | Enhanced extraction torque. | [237] |
| HA, Ti | Enhanced optimal fixation strength. | [238] |
| HA | Enhanced pullout strength. | [65] |
| HA | Enhanced fixation to the bone, reduced the infection and loosening. | [239] |
| HA | Enhanced bone growth. | [240] |
| HA, Ti-6Al-4V | Enhanced bone growth. | [241] |
| HA | Enhanced bone-to-pin interface. | [242] |
| HA | Enhanced bone formation. | [243] |
| Nd-YAG laser | Enhanced surface roughness and maintained fracture resistance. | [44] |

In a study, human fibronectin (FN7-10)-coated MSS was found to enhance bone-implant mechanical fixation and bone contact [224]. Three types of Ti-coated surface-modified MSS (grit blasting, Ti coating, and microarc oxidation) were reported to enhance cell proliferation, alkaline phosphatase activity, migration, and adhesion [57]. A study demonstrated that MSS-based devices coated with polymer poly (L-lactic acid) (PLLA) promoted osteogenesis and bone tissue regeneration and could be a promising coating for supporting bone lesion treatment [225].

A study using bare, HA-coated, and PMMA-cemented MSS implants reported that HA was prone to infection, and the infection developed before HA osteointegration occurred [226]. In a separate study, bulk metallic surface modification of short phosphate glass fibers (sPGF) using ED was demonstrated to be useful for guided bone defect repair [38]. Another promising surface modification of implants is the use of cell adhesion ligands using non-fouling polymer brushes of OEG-methacrylate (MA) on dopamine-functionalized MSS. It was found that the poly(OEG-MA) brushes reduce protein adsorption and cell adhesion, which improve osseointegration, bone growth, and bone regeneration in surgical bone repair [227]. MSS coated with bisphosphonate in a fibrinogen matrix enhanced the pullout strength, and bone remodeling near the implant was reported in a previous study [228]. A comparative study of bisphosphonates and HA coatings suggested that bisphosphonates improved fixation by increasing the amount of surrounding bone, whereas HA mainly improved bone-to-implant attachment [229]. When bisphosphonate was immobilized on MSS, a higher pullout force and pullout energy were observed, suggesting its use for the improvement of biomaterials in bone fixation [230].

A hybrid TEOS-MTES sol–gel-made coating on MSS was reported as a barrier for ion migration and promoter of the bioactivity of the implant surface [71]. An in vivo histomorphological electron microscopic study of HA coating on duplex MSS showed tight apposition between the bone and coating [231]. Investigations of bone healing around uncoated and HA-coated pedicle screws in osteopenic bone demonstrated enhanced bone-

to-implant contact, bone ingrowth, and bone hardness around screws [232]. HA-coated Schanz screws showed no improvement in inhibiting infection but improved the fixation index [233], while a titanium aluminum nitride (TiAlN) coating on MSS resulted in reduced osseointegration between the bone and implant [234]. However, in another study, the HA coating showed a higher bone volume close to the implanted area [235]. In a study, femtosecond laser treatment was used to generate micro-spotted lines on MSS plates, which exhibited improved cell adhesion, bone formation, and decreased fibroblast adhesion. Further investigation is required for real insight into their effectiveness in improving osseointegration and their potential use in clinical applications [236]. HA-coated and uncoated Ti and MSS screws were evaluated to observe osteointegration in ovariectomized cortical bone. The extraction torque for coated screws of both materials was higher than that for uncoated screws, and uncoated Ti had a better extraction torque than uncoated MSS [237].

A study showed that HA-coated screws achieved an optimal fixation strength in the early phase, which was higher than that of standard screws [238]. Similarly, plasma-sprayed HA-coated MSS and Ti rods were stronger than non-coated metals and the pull-out strength of HA-coated Ti was higher than that of HA-coated MSS [65]. HA-coated pins can increase fixation to the bone and reduce the rate of infection and loosening during external fixation for distraction osteogenesis [239]. HA-coated MSS-Schanz screws showed better bone-implant fixation than uncoated screws [240]. Similarly, HA coating was effective in improving the bone-to-pin interface [241,242]. The rate of bone growth in heat-treated and controlled HA-coated metal implants made of Ti alloy (Ti-6Al-4V) and MSS increased the bonding strength of the implants to the host bone for both metals. However, further investigation is required to evaluate this enhanced bone in-growth to metal implants in multiple clinical settings [243]. The biomechanical properties and bone-implant inter-surface response of machined and laser surface-treated MSS mini-screw implants exhibited increased surface roughness without compromising fracture resistance [44].

The methods and findings of surface modification vary in the context of osseointegration. Several aspects of morphological changes in the original MSS and bioactive coatings in MSS were conducted and illustrated by evaluating cell adhesion, proliferation, and bone formation, as well as resistance to bacterial and biofilm growth. At the same time, surface design and roughness are other integral requirements for implant devices that are also taken into account for most of the evaluation process. The modifications are varied and affect different levels with useful features. The review of the literature showed that physical, chemical, and biological modifications in MSS, with intricate biological functions, showed significant effects on the MSS surface modification mechanism. The combination of bioactive, regenerative, cell proliferative, antimicrobial, and growth factor materials that modify the surface character without compromising the unique surface properties of MSS should be continued for future research to develop next-generation MSS that is sustainable and have increased osseointegration and high capacity to reduce bacteria/biofilm growth.

*2.7. Improvement of Physical, Inflammatory, and Miscellaneous Properties*

When designing surface modifications of metal implants, in addition to their biocompatibility, corrosion resistance, bone tissue interface, and protective functionality, it is also important to optimize their mechanical properties, inflammatory responses, and other related factors. Most properties are interrelated. By targeting specific properties, development may lead to other beneficial outcomes. Foreign body acceptance, in terms of biocompatibility and anti-inflammatory responses, of a coated implant device may work. Implant device-host interaction is schematically presented in Figure 8. Table 8 summarizes the improvement in physical, inflammatory, and other miscellaneous properties using several coating strategies on MSS.

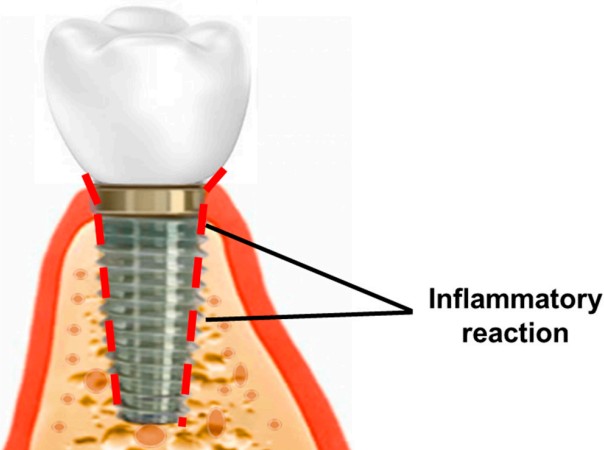

**Figure 8.** Inflammatory response of non-coated implant: Acute/chronic inflammation due to inflammatory products from implants or foreign body reactions; Granulation tissue formation; Fibrosis in soft tissue; Osseo compatibility loss in hard tissue.

In a study, hardystonite bioceramic ($Ca_2ZnSi_2O_7$) was coated onto MSS using ED and showed improved corrosion resistance and bioactivity [39]. A multi-layered coating on MSS was prepared from alternating layers of biocompatible polysaccharide-CH, NSAID, and DCF using electrochemical methods, and the resulting MSS exhibited enhanced corrosion resistance and controlled drug release in a multi-mechanism manner [244]. Sterilization affects the physicochemical properties of coated implants; $SiO_2$ layers were applied to MSS using the sol–gel method, followed by medical sterilization (steam or ethylene oxide). It was demonstrated that the sterilizing agent caused delamination of the layer and decreased the $SiO_2$ barrier properties. However, they did not show any cytotoxicity or negative influence on blood cell counts. Further studies are required to validate these findings [72].

In a study, hydrogenated amorphous carbon (a-C:H) films were deposited using plasma-enhanced CVD on MSS, Ti, and Nb, and were found to reduce the roughness and coefficient of friction while improving the tribomechanical response [28]. The influence of HA on the key physical properties of MSS using electrochemical deposition was also reported [245]. Parylene C was found to have superior mechanical and protective properties as a micrometer-sized coating on MSS [246]. In a multilayer coating, MSS was coated with nano-$TiO_2$ and the endothelial cell-selective adhesion peptide Arg-Glu-Asp-Val and was found to reduce stent restenosis and promote re-endothelialization [247]. Another multicoating was studied using Mg-HA-CT, and it was observed that they formed a uniform surface coating with improved corrosion resistance, bioactivity, and antimicrobial effect without cytotoxicity [248]. The HA-CH dip-coated MSS coating embedded with CA nanofibers exhibited a densely packed uniform film on the substrate, with improved corrosion resistance without cytotoxicity [249]. The deposition of APTS and its co-deposition with propyltrimethoxysilane (PrTMOS) and phenyltrimethoxysilane (PhTMOS) using electrodeposited sol–gel film coating on MSS was reported to be impermeable to some redox species, such as Fe $(CN)_6^{3-/4-}$, and increased the elasticity of the APTS-PhTMOS hybrid [73]. MSS treated with a mixed gas plasma of $NH_3$ and $O_2$ exhibited enhanced resistance to platelet and leukocyte attachment, and in subcutaneous implantation, no inflammation, hemolysis, or untoward thrombosis was observed [250]. Spray-coated poly(2-hydroxy-ethyl-methacrylate) (pHEMA) and a hydrophilic polymeric hydrogel on MSS steel stents were found to deposit firmly on the metal surface and improved roughness, wettability, and morphological and chemical stability [82]. Laser-cut MSS with a coating of polyurethane for a tracheal implant exhibited a holding force similar to that of MSS stents, reduced the histobiological reaction to foreign bodies, and preserved the epithelial structure [251].

Nanocoating of a composite layer of niobium (Nb), tantalum (Ta), and vanadium (V) on MSS using plasma sputtering was reported in a study. Nb and Ta significantly reduced

the friction of the MSS, while V deteriorated the friction of the MMS [252]. The MSS coated with TiN using the PVD method exhibited a decrease in yield strength without structural changes at high temperatures. It was also found to exhibit high resistance to bending stress and friction, as well as sufficient fatigue strength of loading. Additionally, it was reported to be biocompatible [62].

A PDLLA on MSS, Ti6Al4V, and Co-Cr-Mo alloys was found to be stable on the implants and did not influence T-cell reactivity [253]. MSS stents using turbostratic stents showed no clinically relevant reduction in in-stent restenosis and MACE rates compared to uncoated stents [254]. In a study, "G" on MSS surfaces using thermal CVD improved the surface hardness [255]. A further study showed that a drug-incorporated high-methoxyl pectin-xanthan aerogel coating on MSS using ethanol-induced gelation and subsequent supercritical drying exhibited resistance to general corrosion and the release of the drug (NSAID), and this coating was biocompatible with host cells [256]. In a study, MSS was electropolished using an ionic liquid medium based on vitamin B4 and this study revealed the development of smooth nano surface roughness and topography [257]. MSS powders were used with elemental Ni–boron powders and were reported to have high mechanical and corrosion properties as well as biocompatibility [258]. SAMs of long-chain phosphonic acids with $^-CH_3$, $^-COOH$, and $^-OH$ tail groups were created on the native oxide surface of MSS to optimize the interfacial properties and it was found that methyl-terminated phosphonic acid (MTPA) prevented cell adhesion. However, the presentation of hydrophilic tail groups at the interface did not reduce cell adhesion when compared to the control MSS [259]. In a study, graphene sheets were exfoliated directly in a CH solution as a biopolymer, decorated with $TiO_2$ nanoparticles, coated on MSS, and showed improved mechanical properties. No platelet adhesion was observed [260]. The growth of rGO on MSS alloys for biomedical applications was reported in another study. Electrochemical etching increased the concentration of the metal species on the surface and enabled the growth of rGO. In addition, rGO coating did not have toxic effects on mammalian cells [261].

**Table 8.** Surface coating on MSS to improve its physical, inflammatory, and other miscellaneous properties.

| Material (s) | Findings | Refs. |
|---|---|---|
| $Ca_2ZnSi_2O_7$ | Enhanced corrosion resistance and bioactivity. | [39] |
| CH, NSAID, DCF | Enhanced anti-inflammatory response without altering the corrosion resistance and biocompatibility. | [244] |
| $SiO_2$ | Reduced barrier properties and maintained biocompatibility. | [72] |
| a-C:H, Ti, Nb | Reduced the roughness and coefficient of friction, and enhanced tribomechanical properties. | [28] |
| HA | Electrodeposited on an MSS. | [245] |
| Parylene N and C | Enhanced mechanical and protective properties. | [246] |
| Nano-$TiO_2$, Arg-Glu-Asp-Val | Reduced in-stent restenosis and enhanced re-endothelialization. | [247] |
| Mg-doped nano-HA, CH | Enhanced bioactivity, corrosion resistance, and biocompatibility. | [248] |
| HA, CH | Enhanced corrosion resistance, bioactivity, and biocompatibility. | [249] |
| APTS, PrTMOS | Allowed complex geometries coating. | [73] |
| $NH_3$, $O_2$ | Enhanced corrosion resistance, reduced platelet, and leukocyte attachment. | [250] |
| pHEMA | Enhanced interfacial adhesion, withstand shear and tensile stresses. | [82] |
| Polyurethane | Enhanced holding force and reduced histobiological reaction. | [251] |
| TiN | Enhanced fatigue strength and biocompatibility. | [62] |
| PDLLA,Ti-6Al-4V, Co-Cr-Mo | Enhanced mechanical stability without influencing T-cell reactivity. | [253] |
| Carbostent | No reduction of in-stent restenosis. | [254] |
| Graphene | Enhanced surface hardness. | [255] |
| NSAIDs, Pectin, Xanthan | Enhanced corrosion resistance and developed drug-releasing properties. | [256] |
| VB4, Ethylene glycol | Reduced surface roughness without altering the elemental composition. | [257] |
| Boron | Enhanced mechanical properties, corrosion resistance, and bioactivity. | [258] |
| MTPA | Reduced non-specific cell adhesion. | [259] |
| CH, Graphene, Graphene sheets, $TiO_2$NPs | Enhanced the mechanical properties and showed no blood adhesion. | [260] |
| rGO | Enhanced mechanical and biological properties without toxicity. | [261] |
| Nb, Ta, V | Enhanced tribological behavior. | [252] |

### 3. Conclusions

MSS is a corrosion-resistant alloy with a wide range of applications. The passive surface oxide formed in MSS limits metal emissions from the alloy. Although the MSS has a passivating metal oxide layer, the mechanical stability of the MSS is questionable. In aggressive environments such as human physiological fluids and chlorinated environments, it releases nickel and other metals (e.g., cobalt, chromium, copper, iron, and manganese) into physiological fluids, and exhibits toxicity in long-term settings. Therefore, surface modifications and/or morphological functionalization, remodeling, or modification will be beneficial for the use of MMS in the biomedical and dental fields.

Modern research concerned with surface modifications of MSS focuses on cost-effective treatment with optimal outcomes. Surface modification and coating of MSS were documented as one of the main aspects of implant research. Comparing different studies aimed at achieving a modified surface for a specific application is often inspiring and challenging due to differences in the methods for characterizing surface properties. Various materials (i.e., metals, metal oxides, ceramics, and polymers) and methods (i.e., dip coating, electrodeposition, micro-arc oxidation, plasma-spray, sol–gel coating, spin coating, spray coating, etc.) were investigated; however, they are still not standardized. The standardization of tests for assessing the performance of modified MSS is thus obligatory to allow direct comparison between the different methods used for surface modifications. In this review, we figure out different aspects of surface modifications of MSS implants from the perspective of improvement and development of their properties irrespective of their application.

After reviewing a large number of published studies, we concluded that mechanically stable bio-coating is still necessary to overcome this challenge, and sustainable protective coating should be the aim of future research. Although various strategies have useful modifications and their applications, the long-term success and their clinical applications are still unknown. Most of the studies have not advanced to the translational level. Translational and clinical trials should focus on achieving optimal goals rather than conducting only in-vitro-level studies. On the other hand, in this inflationary economy, cost-effective materials should be considered for better outcomes with improved biological functionalization. Therefore, it becomes obvious that future MSS research will succeed if functionality is added using any specific active compound (i.e., biomaterial, antibiotic, antibody, enzyme, etc.) without compromising its key properties, and translating them into clinical settings.

**Author Contributions:** N.S. conducted the conception, design, data acquisition, interpretation, and drafting of the manuscript; Y.N. revised the manuscript; M.Z.I.N. designed and revised the manuscript. All authors have read and agreed to the published version of the manuscript.

**Funding:** The authors received financial support from JSPS (20H05224 and 22H04548).

**Institutional Review Board Statement:** Not applicable.

**Informed Consent Statement:** Not applicable.

**Conflicts of Interest:** The authors declare no conflict of interest.

### Abbreviations

MSS, medical grade stainless steel; ALD, atomic layer deposition; CVD, chemical vapor deposition; ED, electrodeposition; MAPLE, matrix-assisted pulsed laser evaporation; PVD, physical vapor deposition; TEOS, tetraethylorthosilane; MTES, methyltriethoxysilane; HMDSZ, hexamethyldisilazane; PBGHA, poly (lactide-co-glycolide)-bioactive glass-HA; NIPAAm, polymerization of N-isopropylacrylamide; PoP, poly(organo)phosphazene; PC, phosphatidylcholine; PEM, polyelectrolyte multilayer; ASP, active screen plasma.

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
