# Peer review of "Surface Modifications of Medical Grade Stainless Steel"

_coatings, doi:10.3390/coatings14030248_

Round 1
Reviewer 1 Report
Comments and Suggestions for Authors
Dear Authors, thank You for this manuscript. Here below are some recommendations and suggestions
Is this a picture inside the abstract? Or it is a graphical abstract?
The introduction should contain proof of why this topic is urgent. Maybe, a constant rate (a statistical graph) or an increasing trend of the research of MSS. Why this review is important for me.
The methodology of the review also needs more argumentation. Why the keywords were selected in this way?
Also, the article should begin with the analysis and classifications of types of MSS, and methods to improve their bulk properties, deposition of coatings, thermochemical processing, and surface modification methods (without coatings or changes in the surface chemistry). After that, You may start to describe and review these methods.
Also, a good thing in a review article - is to explain its structure in the introduction
Table 1. For each method in the right column give the references (from Your reference list), so the reader can find the source articles You used to review the particular method.
Please, change the order of subsections 2.1-2.7 by Table 1: The order should be the same. Also, table 1 contains 10 basic properties, and in the article, You have only 7 subsections. I could not find the "Bioactivity" subsection. The subtitles quantity and names should be the same, as in Table 1.
Line 360-361. The sentence is unfinished and unclear
Each subsection should have a final summary or mini-conclusion pointing the best methods for improving particular properties, and the limitations, that do not allow to get better results. Or, at least, some general analysis of the used methods. You only give very short information (the "A" did so, and "B" did in another way) from the cited articles, with no results. How can the reader compare the cited results?
After section 2, before the conclusions, there should be a kind of discussion, which results in new research ways to overcome the limitations discovered in the article based on the literature review. This section is missing.
The conclusions also should be rewritten, as far as they do not highlight or argument any idea, accept "we have to work and then we will get a better result".

Reviewer 2 Report
Comments and Suggestions for Authors
The manuscript submitted for review provides a comprehensive review of the literature on surface modification of medical-grade stainless steel. The manuscript meets the journal's requirements, so I recommend publishing the manuscript after reading the comments below.
The introduction suggests a detailed description of the chemical and mechanical properties of the steel that is the subject of this manuscript. It would be good to also present this information in tabular form. This way of writing will improve the readability of the article.
At the beginning of the article, I also propose to present in detail the use of steel. In this case, a graphical notation would also be appropriate.
Table 1 is not the best solution to present the surface modifications of the steels in question. It is proposed to create appropriate graphics (not necessarily tables) for steel modification techniques and separate graphics for target properties.
The content of the article should precisely show the relationship between the technique used and the impact on improving the properties of steel.
When reading the manuscript, one has the impression that not all of the target properties indicated in Table 1 are described later in the manuscript.
The order in which target factors should be discussed should be in relation to the list of factors shown in Table 1.
Reviewer 3 Report
Comments and Suggestions for Authors
This a good review paper. The author review the surface modifications of stainless steel and its application on medical.
I only have two comments, (1)Table 1, please add the reference on the every modification techniques and target properties . (2) Table 1, for UV irradiation, I am not sure it is a effective method to improve the properties of SS, Long-term Effectiveness?
Reviewer 4 Report
Comments and Suggestions for Authors
The objective of the manuscript, according to the authors is "This comprehensive review summarizes recent advances in the surface modification of MSS-based implants and their applications in the biomedical field."
#1 Considering the "recent advances" the list of references, clearly show that it is not the case. In 244 references ONLY 4 are from 2020 and 2 from 2021. All other are from previous decades and century. A manuscript that intends to show the recent advances must present, at least 75% of the references from the last 3 or 4 years!!
#2 - The introduction is very short and very poor in scientific content.
#3 - The authors indicate applications in the biomedical field, but only relevance is given to Dentistry.
#4 - In implants that contact hard tissue (bone) such as in Dentistry, are more prompt to wear and release of particles and ions. For this reason the most used alloy is not MSS but Ti-based alloys. The reason is that MSS has in its chemical composition relative high amounts of Ni, so they do not present magnetic properties. Ni will open the austenitic domain of the steel until room temperature and austenitic steel are not magnetic. The real problem is that the release of Ni constitutes a serious and deadly problem to 1/3 of the World population that is allergic to this metals and its ions.
Due to the problematic of scientific issues presented in this manuscript I do not think that is appropriate to be published in its current version.
Round 2
Reviewer 1 Report
Comments and Suggestions for Authors
Dear Authors, thank You for the work done. Good luck!
Author Response
Dear Reviewer,
Thank you so much for your appreciation.
Sincerely,
Mohammed Zahedul Islam Nizami
Reviewer 4 Report
Comments and Suggestions for Authors
I apologize to the authors that make an effort trying to improve the manuscript.
However, the major scientific concerns that I exposed in the first revision are still there. Moreover, when the authors state in their response that that are very few published literature on the subject within the last years, it is a "red flag" on the assumption that is an exciting topic. In fact is a topic that as been showing less and less interest in the biomaterial field.
Therefore, my final decision continues to be to reject the submission.
